# Neural circuitry of a polycystin-mediated hydrodynamic startle response for predator avoidance

Luis A Bezares-Calderón[1,2], Jürgen Berger[2], Sanja Jasek[1,2], Csaba Verasztó[1,2], Sara Mendes[2], Martin Gühmann[2], Rodrigo Almeda[3], Réza Shahidi[1,2], Gáspár Jékely[1,2]*

[1]Living Systems Institute, University of Exeter, Exeter, United Kingdom; [2]Max Planck Institute for Developmental Biology, Tübingen, Germany; [3]Centre for Ocean Life, Technical University of Denmark, Denmark, Kingdom of Denmark

**Abstract** Startle responses triggered by aversive stimuli including predators are widespread across animals. These coordinated whole-body actions require the rapid and simultaneous activation of a large number of muscles. Here we study a startle response in a planktonic larva to understand the whole-body circuit implementation of the behaviour. Upon encountering water vibrations, larvae of the annelid *Platynereis* close their locomotor cilia and simultaneously raise the parapodia. The response is mediated by collar receptor neurons expressing the polycystins PKD1-1 and PKD2-1. CRISPR-generated *PKD1-1* and *PKD2-1* mutant larvae do not startle and fall prey to a copepod predator at a higher rate. Reconstruction of the whole-body connectome of the collar-receptor-cell circuitry revealed converging feedforward circuits to the ciliary bands and muscles. The wiring diagram suggests circuit mechanisms for the intersegmental and left-right coordination of the response. Our results reveal how polycystin-mediated mechanosensation can trigger a coordinated whole-body effector response involved in predator avoidance.
DOI: https://doi.org/10.7554/eLife.36262.001

*For correspondence:
g.jekely@exeter.ac.uk

## Introduction

Approaching predators or other threatening stimuli often elicit prey escape or startle responses characterized by rapid whole-body locomotory actions (*Bullock, 1984*). For example, crayfish escape from threatening stimuli by rapid tail flips (*Edwards et al., 1999*) and fish perform a rapid 'C-shape' body bend upon vibrational or visual stimuli (*Eaton et al., 1977*). The ubiquity and stereotypy of such responses have allowed their study in a variety of organisms, uncovering many commonalities in the underlying neuronal circuitry.

Startle responses require the simultaneous activation of a large number of muscles in a coordinated manner with a short latency (*Eaton et al., 1977*). This is often achieved by a system of giant command neurons or motoneuron such as the giant fibres in crayfish (*Wiersma and Ikeda, 1964*), the Mauthner cells (M cells) in fish and lampreys (*Korn and Faber, 2005*) or the giant motor axons in the jellyfish *Aglantha digitale* (*Roberts and Mackie, 1980*). Such giant neurons increase transmission speed and accuracy, and enable activation of the locomotor system with the least number of intervening synapses (*Bullock, 1984*).

Another common feature of startle circuits is the presence of converging neuronal pathways that activate the same effectors. This can contribute to behavioural flexibility. In crayfish, a non-giant circuitry is present that enables the animal to respond more variably to less abrupt stimuli (*Edwards et al., 1999*; *Kramer et al., 1981*). In fish, not only the M cell but also other descending reticulospinal neurons are required for short latency responses (*Liu and Fetcho, 1999*). Convergence

can also occur at the sensory level, as shown in fish, crayfish and *Drosophila* larvae (*Lacoste et al., 2015*; *Ohyama et al., 2015*; *Zucker, 1972*).

The motor programs during a startle response can also differ depending on the location of the stimulus. This is due to differences in connectivity downstream of different sensory fields. In *C. elegans* (*Chalfie et al., 1985*) and crayfish (*Wine and Krasne, 1972*), partially overlapping circuits induce forward movement upon posterior stimulation, or backward movement upon anterior stimulation. In *Drosophila* larvae, backward or forward movement depends on stimulus direction. This is achieved by an interneuron type with differential connectivity in anterior and posterior segments (*Takagi et al., 2017*).

Startle responses and their flexibility can best be interpreted by considering the ethological context. For example, in *C. elegans*, defects in the touch response lead to a lower escape rate from fungal predators (*Maguire et al., 2011*). Crayfish display giant-fibre-mediated tail flips to avoid approaching predators, but non-giant-mediated tail flips to escape following capture (*Herberholz et al., 2004*). Studying startle behaviours using naturalistic stimuli could thus help reveal the behavioural features that have been selected for by evolution.

Startle responses are useful models to uncover the relationship between genes, circuits and behaviour. The first organism where behavioural, genetic and whole-body connectomics analysis could be integrated was the nematode *C. elegans*. Here the study of the touch withdrawal response led to the discovery of key components of the mechanosensory cascade including the DEG/ENaC mechanotransduction channel complex MEC-4 (*O'Hagan et al., 2005*)(reviewed in *Schafer, 2015*). Genetic and cell ablation techniques, combined with the analysis of the whole-body wiring diagram (*White et al., 1986*) allowed the characterisation of the sensory-locomotor circuits involved in the touch withdrawal response in *C. elegans* (*Chalfie et al., 1985*; *Piggott et al., 2011*).

Multilevel studies integrating whole-body connectomics, behaviour, genetics and ethology in a larger diversity of animals will help to identify general principles of the neural control of behaviour. A broader comparative approach will also be important to understand how circuits and behaviours have evolved. Here we combine behavioural assays, calcium imaging, genetic analysis and circuit reconstructions to characterise a hydrodynamic startle response in larval *Platynereis dumerilii*, a marine annelid accessible for whole-body connectomics, genetic manipulations and behaviour analysis (*Randel et al., 2015*; *Randel et al., 2014*; *Verasztó et al., 2017*; *Williams and Jékely, 2016*; *Zantke et al., 2014*).

## Results

### A hydrodynamic startle response in *Platynereis* larvae

Nectochaete larvae of the marine annelid *Platynereis dumerilii* are planktonic and swim with beating cilia arranged into segmental bands (trochs) (*Figure 1A*). The larva has three main trunk segments, each with two pairs of parapodia endowed with spiny chaetae and a complex musculature (*Fischer et al., 2010*). Upon disturbances to the surrounding water, freely swimming larvae abruptly stop swimming, contract the body and simultaneously elevate parapodia in all segments and on both sides of the body (*Figure 1B–C*). This response is followed by a slower recovery phase (*Figure 1—figure supplement 1*, *Video 1*).

To characterize the kinematics of the startle response, we recorded trunk-tethered larvae at 230–350 frames per second during fictive swimming and stimulated them with a vibrating tungsten filament placed at a distance from the larvae to trigger startle responses (*Figure 1D* and *Video 2*). We visualized the water flow with fluorescent microbeads (*Video 3*). When we stimulated larvae from the anterior, they closed their beating cilia and elevated the parapodia. Both aspects of the response were modulated as a function of filament speed (*Figure 1G*). The main ciliary band (prototroch) closed in all trials except at the lowest filament speeds tested (*Figure 1G*). The closures extended to all locomotor ciliary bands (hereafter referred to as bodytrochs) (*Video 2*). Although parapodial elevation was triggered with some of the lowest filament speeds tested, it was only consistently observed above 28 µm/ms. The extent of the elevation of parapodia was dependent on the speed of the filament and displayed a bimodal distribution (*Figure 1G*). We split the distribution into low-angle and wide-angle elevation responses (LowE and WideE, respectively) by a cutoff obtained from adjusting the data to a finite mixture model (*Figure 1E–G*, *Video 2*). LowE responses

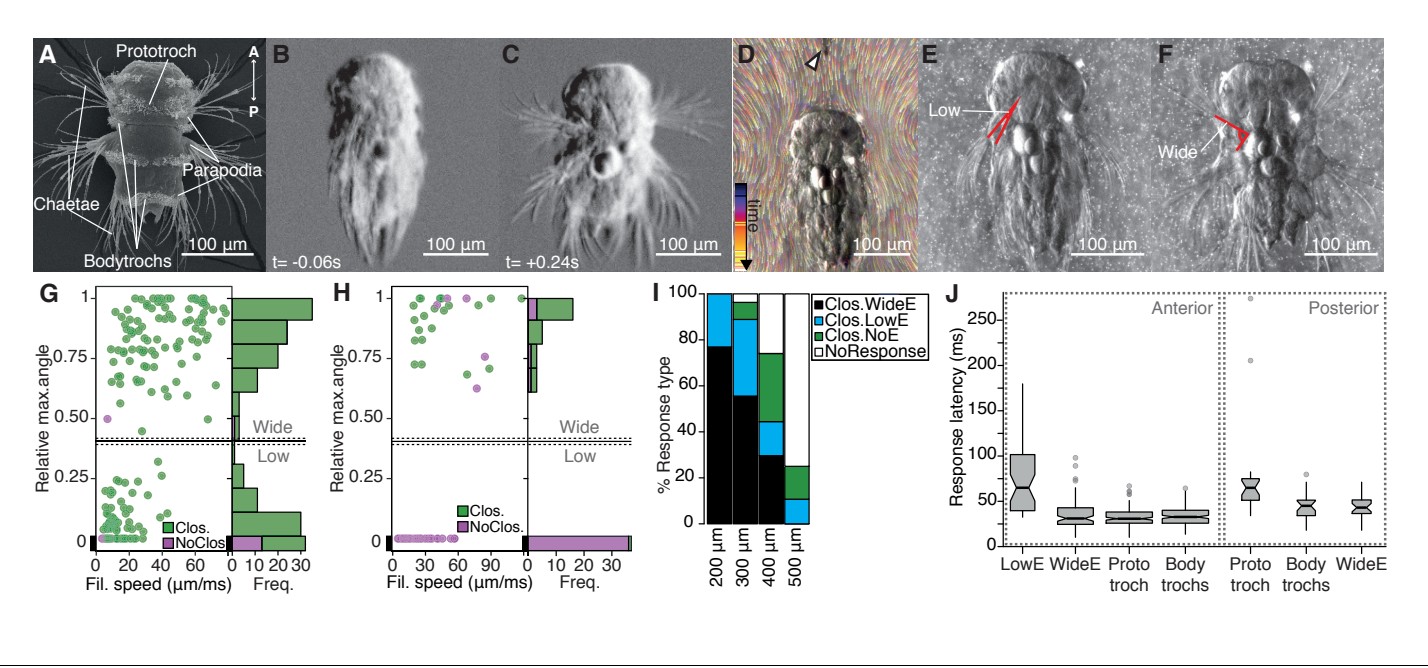

**Figure 1.** The startle response in *Platynereis* larvae is driven by hydrodynamic disturbances. (A) SEM micrograph of a 3-day-old nectochaete larva, dorsal view. (B–C) Snapshots of *Video 1* 60 ms before (B) and 240 ms after (C) a vibration stimulus. (D) Trunk-tethered larva (arrowhead points to the filament used for stimulation) engaging in (fictive) swimming. Fluorescent beads are colour coded by frame. (E–F) Examples of low-angle (E) or wide-angle (F) parapodial elevation upon anterior stimulation (angles measured are outlined in red). (G–H) Relative parapodial elevation angle (0, no elevation; 1, maximum elevation) as a function of filament speed. The stimulation filament was placed 100 μm from the head (G) or pygidium (H). Dots are colour coded by ciliary band (prototroch) closure. A stacked histogram is shown to the right of each scatter plot. A finite Mixture Model-based cut off (0.4, solid horizontal lines) splits the two main populations of the histogram in G. Dashed horizontal lines: 95% CI. (I) Stacked bar plot of startle response profiles upon stimulation with a filament placed at varying distances from the head. (J) Latency distributions for the onset of each response upon anterior (left) or posterior (right) stimulation. Notch in boxplots displays the 95% CI around the median. Box width proportional to $\sqrt{n}$. n = 9 larvae in panels G) to J), each tested multiple times.

DOI: https://doi.org/10.7554/eLife.36262.002

The following source data and figure supplements are available for figure 1:

**Source data 1.** Source data of startle kinematics experiments.

DOI: https://doi.org/10.7554/eLife.36262.005

**Figure supplement 1.** Descriptive statistics of the startle response in freely-swimming and tethered larvae.

DOI: https://doi.org/10.7554/eLife.36262.003

**Figure supplement 1—source data 1.** Normalized speed and area values of startle responses in freely swimming larvae.

DOI: https://doi.org/10.7554/eLife.36262.004

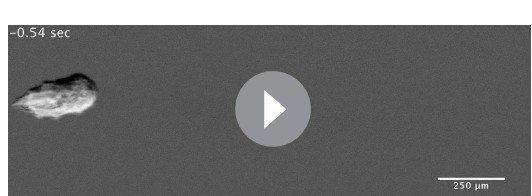

**Video 1.** Startle response of *Platynereis* larvae. Time relative to start of stimulus.

DOI: https://doi.org/10.7554/eLife.36262.006

were more common at low to moderate speeds, while WideE responses were seen across a broad stimulus-intensity spectrum (*Figure 1G*, *Figure 1—figure supplement 1*). WideE responses had a shorter latency and the maximum parapodial elevation was achieved sooner than LowE responses (*Figure 1J*, and data not shown). Ciliary arrests and WideE responses occurred in close succession to each other (*Figure 1—figure supplement 1*). As we moved the filament further away from the head, the response profile shifted to greater filament speeds (*Figure 1I*), showing that the response is triggered by near-field hydrodynamic disturbances.

Next, we stimulated larvae from the posterior end (pygidium) from a 100 μm distance. Posterior stimulation triggered startle behaviours with a markedly different response profile compared

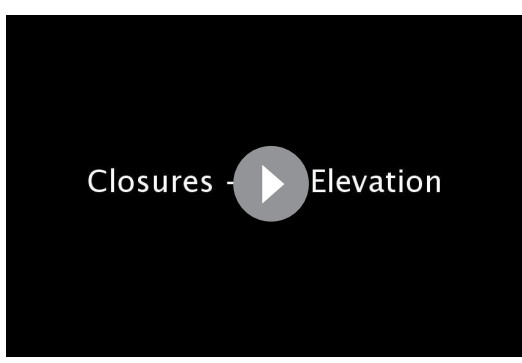

**Video 2.** Graded startle responses of a glued *Platynereis* larva stimulated with a vibrating filament. Low stimulus intensity triggers ciliary closures only (as assessed by stop of the fluorescent beads around the larva), intermediate stimulation induces ciliary closures and low-angle parapodial elevation, and high intensity stimulation induces ciliary closures and wide-angle parapodial elevation (WideE). Time is relative to stimulus start.

DOI: https://doi.org/10.7554/eLife.36262.007

to anterior stimuli (*Figure 1H*). First, animals were less sensitive to posterior stimulation as only filament speeds >19 µm/ms triggered a response. Second, we only observed WideE and no LowE responses. The WideE responses were often not accompanied by ciliary closures. On the contrary, we only very rarely observed (prototroch) ciliary closures not accompanied by a WideE response. WideE responses triggered from anterior and posterior stimulation were similarly fast (anterior and posterior WideE response median latency: 31.7 ms and 42 ms, respectively) (*Figure 1—figure supplement 1*) WideE responses accompanied by prototroch ciliary closures were temporarily uncoupled (ciliary closures median latency: 65 ms (prototroch) or 45.6 ms (bodytrochs);) (Figure 1J, K).

During WideE responses the three chaeta-bearing segments and the left and right body sides were highly coordinated. As already apparent from freely swimming larvae (*Figure 1C*) the extent of the elevation was bilaterally symmetric in all responses observed (*Figure 1F* shows a representative example). The delay values between the onset of elevation of the first pair of left and right parapodia were mostly below our recording speed limit (*Figure 1—figure supplement 1*). Parapodia in different segments, but on the same body side rose simultaneously or one frame apart from each other (*Figure 1—figure supplement 1*).

We thus characterized a rapid, left-right symmetric, segmentally coordinated, stereotypical whole-body startle response. The response is triggered by hydrodynamic stimuli and has a scalable profile that is different depending on the direction of stimulus.

## Collar-receptor neurons respond to water-borne vibrations

To understand the neuronal mechanism of the startle response, we searched for candidate mechanosensory neurons in *Platynereis* larvae. The most likely candidates to sense flow-driven strain and bending are neurons with ciliary structures penetrating the cuticular surface (*Budelmann, 1989*). Using a combination of scanning (SEM) and ssTEM, we identified and mapped all penetrating uniciliated, biciliated and muliciliated sensory neurons across the whole body of nectochaete larvae (*Figure 2* and *Figure 2—figure supplement 1*). Since nectochaete larvae have a stereotypical morphology, the same cells could be reliably identified in different larvae by SEM and in the ssTEM dataset (*Randel et al., 2015*).

We focused on penetrating ciliated sensory neurons on the head of the animal, as this region was the most sensitive to mechanosensory stimuli (*Figure 1*). We found different candidate cell types (*Figure 2A–B*), including a group of cells very similar to the collar receptor cells (CRs)

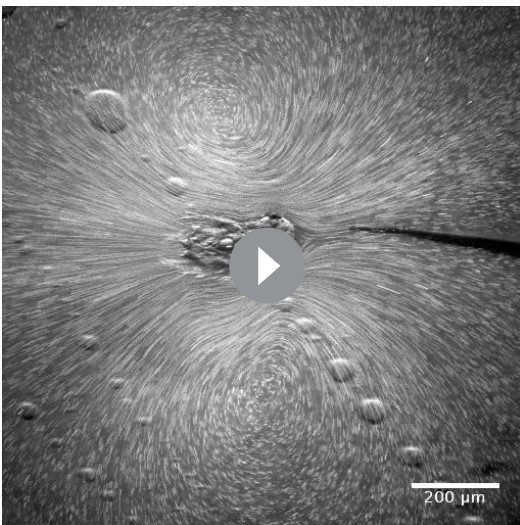

**Video 3.** Flow field generated by beating cilia around a glued *Platynereis* larva. Fluorescent beads were added in order to visualize flow. The stimulation needle is located in front of the larva. Some drops of dried glue are visible.

DOI: https://doi.org/10.7554/eLife.36262.008

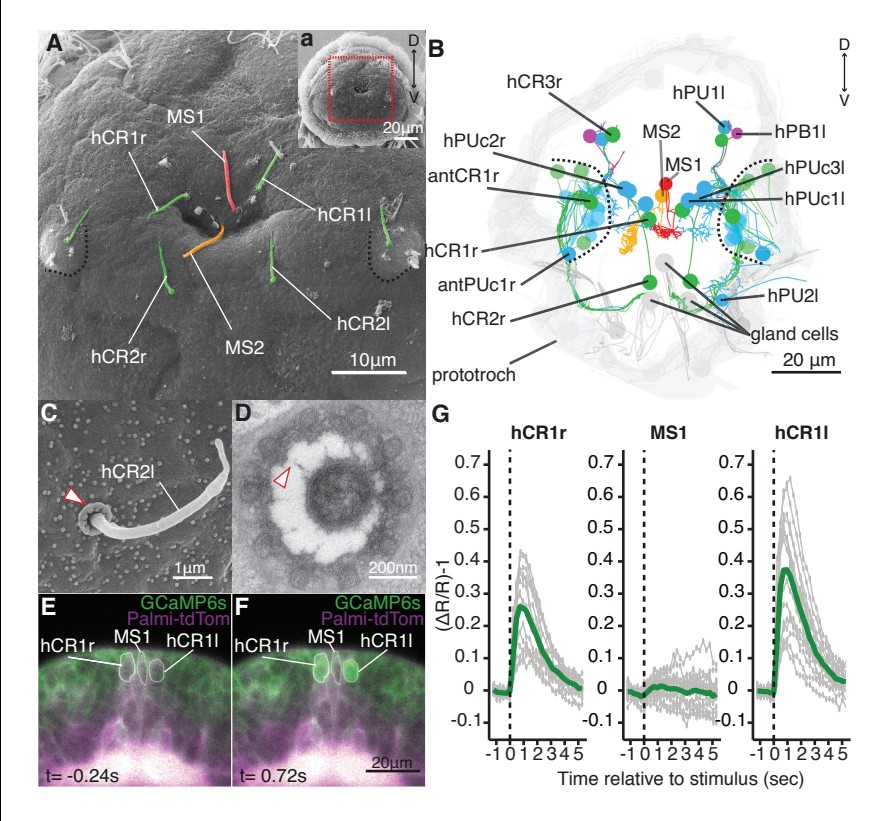

**Figure 2.** Head CR neurons are hydrodynamic receptors. (**A**) Close-up view of the head of a nectochaete larva (region outlined in a). Developing antennae with a CR cilium (green) are outlined in blue. (**B**) Reconstruction of penetrating ciliated sensory neurons in the head from a serial transmission electron microscopy volume. Colour code as in (**A**). Additional penetrating uniciliated (PUc and PU) or biciliated (PB) neurons are shown in blue and magenta, respectively. One of each bilateral pair is labelled. Cells not fully penetrating the cuticle are shown in semi-transparent colors. (**C**) SEM of a CR sensory cilium (hCR2l), collar of microvilli protrudes out of the cuticle (arrowhead). (**D**) TEM cross-section of a CR sensory cilium with a collar of 10 thick microvilli and fibres connecting the cilium and the microvilli (arrowhead). (**E, F**) Snapshots from *Video 5* showing a larva injected with *Palmi-tdTomato-P2A-GCaMP6s* mRNA 0.24 s prior (**E**) or 0.72 s after (**F**) stimulation with a vibrating filament (ventral view). The ROIs used for fluorescence quantification are outlined in white. (**G**) Mean ratiometric fluorescence changes (green traces) of hCR1l, hCR1r and MS1 neurons upon stimulation with a vibrating filament placed 50 µm anterior to the head. 15 (hCR1r), 18 (MS1) and 16 (hCR1l) measurements (grey traces) from five animals are shown. The traces were aligned relative to the stimulus start (t = 0).

DOI: https://doi.org/10.7554/eLife.36262.009

The following source data and figure supplements are available for figure 2:

**Source data 1.** Source data of calcium imaging experiments.
DOI: https://doi.org/10.7554/eLife.36262.012
**Source data 2.** Sequence and map of plasmid construct used to synthesize Palmi-tdTomato-P2A-GCaMP6s mRNA.
DOI: https://doi.org/10.7554/eLife.36262.013
**Figure supplement 1.** Penetrating ciliated sensory neurons in the nectochaete larva.
DOI: https://doi.org/10.7554/eLife.36262.010
**Figure supplement 2.** Distinct head CR and MS neurons are identifiable in living animals.
DOI: https://doi.org/10.7554/eLife.36262.011

previously identified in other polychaetes (*Budelmann, 1989*; *Purschke, 2005*; *Purschke et al., 2017*; *Schlawny et al., 1991*; *Windoffer and Westheide, 1988*), earthworms (*Knapp and Mill, 1971*) and leeches (*Phillips and Friesen, 1982*). The CR neurons in *Platynereis* have a single non-motile cilium with a $9 \times 2 + 2$ microtubule doublet pattern and a symmetric collar of 10 thick microvilli surrounding the cilium (*Figure 2C–D*). Each microvillus has a dense region in the inner side and

is connected by thin fibres to the cilium (*Figure 2D*). The collar and the cilium penetrate the cuticle, thus these cells can also be identified in SEM samples (*Figure 2C*). CR neurons also occur in other regions of the nectochaete larva, either individually or in clusters as parts of developing organs (*Figure 2A,B*, and *Figure 2—figure supplement 1*).

To determine if the head CR neurons are hydrodynamic mechanoreceptors, we recorded the activity of a set of clearly identifiable head CR neurons (hCR1 and hCR2) by calcium imaging (*Figure 2E*, *Figure 2—figure supplement 2* and *Video 4*). We also imaged another distinct type of collared uniciliated penetrating neuron called MS1 located between the two hCR1 cells (*Figure 2A–B* and *Figure 2—figure supplement 2*). The cell bodies of hCR1 and MS1 neurons lie in the same focal plane and could be recorded simultaneously (*Figure 2E,F*, *Figure 2—figure supplement 2*). When we stimulated larvae with a vibrating filament, the cilia of the hCR1 cells deflected and GCaMP6s fluorescence increased in the cell bodies and in the cilia (*Figure 2E–G*, *Video 5* and *Video 6*). Although the cilium of MS1 was also deflected, the fluorescence did not increase in this cell (*Figure 2G*, and *Video 6*). hCR2 neurons were also activated by the stimulus (*Figure 2—figure supplement 2*). These results show that head CR neurons detect hydrodynamic vibrations and could thus trigger the startle response.

## CR neurons express polycystin channels

To investigate the mechanism of hydrodynamic reception, we searched for mechanosensory markers specifically expressed in CR neurons. We found that a *Platynereis* ortholog of the TRPP/Polycystin-2 channel, *PKD2-1*, (*Figure 3—figure supplement 1*) was expressed in CR neurons, as demonstrated by *in situ* hybridization and a transgenic reporter construct (*Figure 3A,B,D*). *PKD2-1* was also expressed in two other types of penetrating uni- and biciliated sensory neurons (PU and PB neurons, respectively) in the head, trunk, and pygidium, including an unpaired biciliated sensory cell in the pygidium (pygPB[unp]) (formerly pygPBbicil;*Shahidi et al., 2015*; *Verasztó et al., 2017*)(*Figure 3—figure supplement 2*). Another gene in *Platynereis*, *PKD1-1*, was expressed only in CR neurons and in the pygPB[unp]neuron at the nectochaete stage (*Figure 3A,C,E* and *Figure 3—figure supplement 3* and *Video 7*). *PKD1-1* belongs to the TRPP-related family PKD1, and it defines a novel invertebrate family that is paralogous to the more widespread Polycystin-1 and PKD1L1 families (*Figure 3—figure supplement 1*).

## *PKD1-1* and *PKD2-1* are required for the startle response

The co-expression of *PKD1-1* and *PKD2-1* in CR neurons and the known function of their homologs in mechanotransduction (*Nauli et al., 2003*; *Sharif-Naeini et al., 2009*) suggested that polycystins participate in the mechanotransduction cascade in CR neurons. To test this, we generated mutant lines with the CRISPR/Cas-9 system (*Lin et al., 2014*). We recovered multiple deletion alleles in both *PKD1-1* and *PKD2-1* with frameshift mutations and premature STOP codons leading to protein products predicted to lack most functional domains (*Figure 4A–D*). Homozygote and trans-heterozygote larvae for either *PKD1-1* or *PKD2-1* alleles were viable and fertile and had normal CR neuron morphology (*Figure 4—figure supplement 1*). These larvae, however, failed to startle upon touching their head, a stimulus that triggered the startle response in most wild type and heterozygote larvae (*Figure 4E–F*, *Figure 4—source data 1*). We also tested the responses of tethered *PKD1-1* and *PKD2-1* mutant larvae stimulated from the anterior or the posterior end with a vibrating filament (*Figure 4G–H*, *Video 8*, *Figure 4—source*

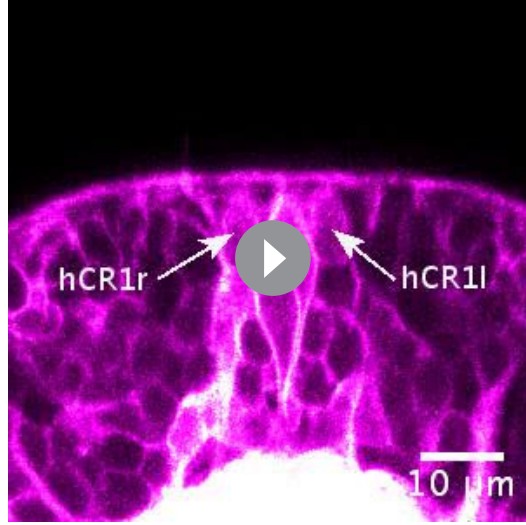

**Video 4.** Morphology of the hCR1 neurons marked with Palmi-tdTomato. Circles track the dendrite projection towards the cilia of each hCR1.
DOI: https://doi.org/10.7554/eLife.36262.014

*data 2*). All aspects of the startle response, including ciliary arrests and parapodial extensions were absent from mutant larvae for either of the genes across the whole range of filament speeds tested. This demonstrates that the neuronal expression of *PKD1-1* and *PKD2-1* channels is essential for the hydrodynamic startle response in *Platynereis* larvae.

The absence of a startle response in the otherwise normal polycystin mutants allowed us to test whether the startle response plays a role in prey-predator interactions, as suggested for other planktonic larvae that have a similar response (*Pennington and CHIA, 1984*; *Thiel et al., 2017*). We exposed *Platynereis* larvae to *Centropages typicus*, a predatory copepod which detects its prey by the hydromechanical signals generated by prey locomotion (*Calbet et al., 2007*; *Cowles and Strickier, 1983*; *Kerfoot, 1978*) (*Figure 5A*). We incubated in the same container an equal amount of *PKD2-1^{mut/mut}* and age-matched wildtype larvae with *C. typicus* and quantified the number of survivors of each genotype after 12 hr or 24 hr. We also incubated larvae without copepods to control for differences in mortality rates not related to predation (*Figure 5—source data 1*). In most experiments, *PKD2-1^{mut/mut}* larvae were predated more than their wildtype counterparts (p < 0.001, one-sided exact Wilcoxon-Pratt signed rank test, *Figure 5B*). As wild type but not mutant larvae showed the startle response upon copepod attacks (*Video 9*), the difference in predation rates is likely due to the absence of a startle response in *PKD2-1^{mut/mut}* larvae.

## Whole-body wiring diagram of the startle response

The hydrodynamic sensitivity of the head CR neurons (*Figure 2*), their expression of *PKD* genes (*Figure 3*), and the requirement of these genes for the startle response (*Figure 4*) argue for the CRs as the main sensory receptors that trigger the startle response. To investigate the neuronal mechanisms behind the response, we mapped the wiring diagram of chemical synapses downstream of all CRs in a whole-body ssTEM dataset (*Randel et al., 2015*). We identified all the direct postsynaptic partners (more than two synapses from more than 1 CRs) of CRs in the larva (*Figure 6—source data 1–2*, *Video 10*). We only analysed the most direct neural paths from CRs to the ciliated and muscle effectors, assuming the initiation of the response here described would use the path with the fewest intervening synapses. The head and pygidial CRs project along the ventral nerve cord (VNC) posteriorly and anteriorly, respectively (*Figure 6A* and *Figure 6—figure supplement 1*), and form *en passant* synapses with distinct types of interneurons (INs) and motoneurons (MNs) (*Figure 6B–F*, *Figure 6—figure supplement 1*). These INs in turn target muscle-motor and ciliomotor neurons that innervate several muscle types and the ciliary bands in the larva on both body sides (*Figures 6–8*).

The head CRs directly synapse on the MC cell, a previously described cholinergic motoneuron that induces ciliary closures of the prototroch ciliary band (*Verasztó et al., 2017*) (*Figure 6B*). The head but not the pygidial CRs also synapse extensively on CM interneurons (IN^{CM}), a bilateral pair of ipsilaterally projecting pseudo-unipolar neurons with a soma in the 1^{st} segment (*Figure 6B–C*). The CMs are presynaptic to the Loop neurons on the same side and to MN^{ant} on both body sides (*Figure 6B* and *Figure 8*). Loop and MN^{ant} are ciliomotor neurons that likely control the closures of prototroch and bodytroch cilia (*Verasztó et al., 2017*)(*Figure 6B*). Head CRs also synapse on a novel interneuron type, the Rope interneurons (IN^{rope}). These neurons have their soma in the head and have ipsilateral descending projections that span the entire VNC (*Figure 6C*). IN^{rope} neurons make synaptic contacts with the Loop neurons (*Figure 6B*), thus providing a converging pathway to ciliary control. Finally, both head and pygidial CRs target the sensory-motor neuron pygPB^{unp}, which may modulate the prototroch ciliary band system (*Verasztó et al., 2017*).

CRs also feed into distinct muscle-motor pathways (*Figure 6B*). Both head and pygidial CRs directly synapse on the Crab motoneurons (MN^{crab}), a unique set of decussating VNC motoneurons that innervate a variety of trunk muscles (*Figure 6B*, *Figure 6—figure supplements 1* and *2*). A second pathway is through the IN^{rope} interneurons (*Figure 6B*). The descending projections of Rope neurons target several segmentally arranged ipsi- and contralaterally projecting VNC motoneurons (MN^{vnc}) that innervate distinct sets of somatic muscles (longitudinal, oblique, transverse, and parapodial groups) in all three segments and on both body sides (*Figure 7A–C*, *Figure 8*, *Figure 6—figure supplements 1* and *2*).

An alternative muscle-motor pathway is through the Split interneurons (IN^{split}), a group of segmentally arranged ipsilateral pseudounipolar cells (similar in morphology to CMs) (*Figure 6D* and *Figure 6—figure supplement 1*,*Video 10*). IN^{split} neurons synapse onto numerous ipsi- and contralaterally projecting MN^{vnc} neurons in all segments (*Figure 7A* and *Figure 8*), some of these MNs are

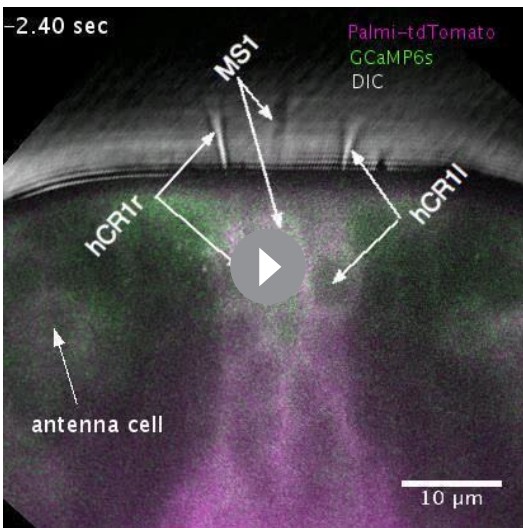

**Video 5.** Calcium imaging from the hCR1 and MS1 neurons during a vibrational stimulation. The larva is ubiquitously expressing Palmi-tdTomato and GCaMP6s. Time is relative to stimulus start.
DOI: https://doi.org/10.7554/eLife.36262.015

also targeted by IN^rope (see Venn diagram in *Figure 7A*; *Figure 6—figure supplement 2A*). IN^split cells are postsynaptic to head, trunk and pygidial CRs (*Figure 6B*).

CRs also synapse on a heterogenous group of ascending and descending trunk commissural interneurons (IN^comm) (*Figure 6E* and *Figure 6—figure supplement 1*). In contrast to the other interneuron types, we could not find bilateral pairs of IN^comm targeted by left and right CRs (*Figure 6—figure supplement 1*). Some of these commissural neurons synapse on contralateral interneurons and motoneurons in the startle network and also feed back to the CRs (*Figure 6B* and *Figure 8*). The majority of motoneurons targeted by IN^comm are also targeted by IN^split and IN^rope (see Venn diagram in *Figure 7A*).

The VNC motoneurons targeted by IN^split, IN^rope and IN^comm belong to at least ten morphologically distinct groups, with segmentally iterated and left-right symmetric members (*Figure 7A,B*, *Figure 8*, *Figure 6—figure supplement 2*). The contralateral motoneurons MN^crab, MN^ring, MN^spider and MN^smile innervate all the distinct muscle classes involved in the startle response (longitudinal, oblique, transverse, axochord and parapodial groups) and account for most of the synaptic targets on muscles. Ipsilateral neurons target exclusively oblique and parapodial muscles (*Figure 8*, *Figure 6—figure supplement 2*). This is also true for MN^bow, a motoneuron class with both ipsilateral and decussating axonal projections *Figure 6—figure supplement 2*). By calcium imaging during startle responses, we could directly confirm the contraction of the longitudinal, oblique, axochord and parapodial muscle groups (*Figure 7C* and *Video 11*).

# Discussion

**Video 6.** Calcium imaging from the hCR1 and MS1 neurons during a vibrational stimulation. The stimulation filament and the deflection of the sensory cilia are visible. The larva is ubiquitously expressing Palmi-tdTomato and GCaMP6s.
DOI: https://doi.org/10.7554/eLife.36262.016

## Collar receptor neurons mediate a hydrodynamic startle response in *Platynereis* larvae

In this study, we described a startle behaviour in the planktonic *Platynereis* larva triggered by hydromechanical disturbances. We identified a group of mechanosensory neurons, the CRs, that respond to hydrodynamic stimuli and express the polycystin homologs *PKD1-1* and *PKD2-1*. Through CRISPR mutagenesis, we showed that the startle response requires both *PKD1-1* and *PKD2-1*. We also provided evidence for the importance of *PKD2-1* in escaping a rheotactic predator. Connectomic analysis of the CR sensory-motor circuit identified distinct circuit motifs that could explain the whole-body temporal and spatial coordination characteristic of the startle behaviour (*Figure 8*; *Supplementary file 1*).

We identified the CR neurons as the hydrodynamic mechanosensory receptors initiating the startle response. CRs have a sensory dendrite with a cilium and a collar of microvilli, an arrangement commonly found in mechanoreceptors in other annelids (*Budelmann, 1989*; *Knapp and Mill, 1971*; *Phillips and Friesen,*

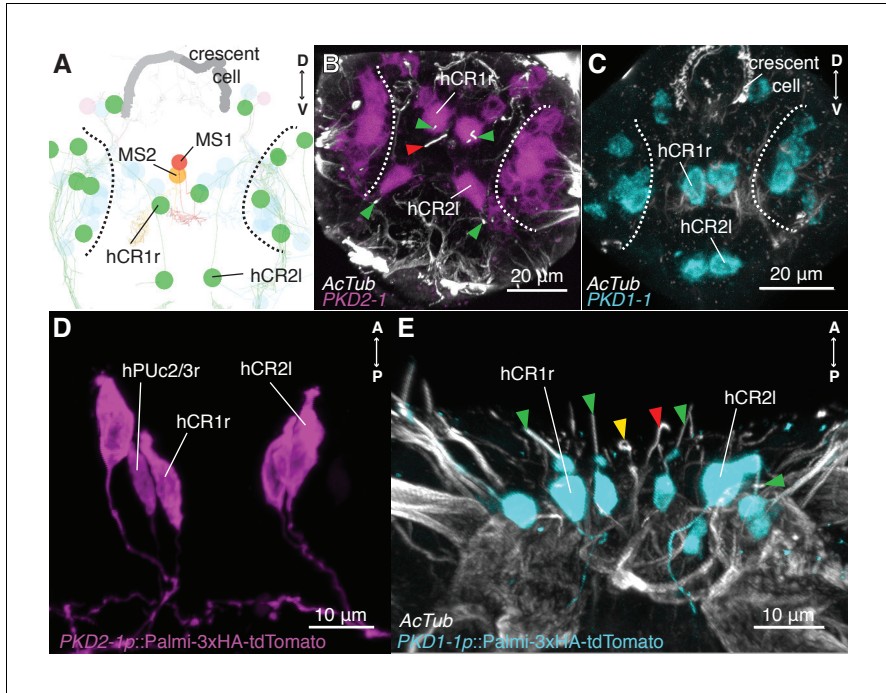

**Figure 3.** CR neurons express *PKD1-1* and *PKD2-1*. (**A**) Electron microscopy volume reconstruction of penetrating ciliated cells in the head. Only soma position of CRs (green) and MS cells are highlighted for comparison with panels B and C. Black dashed lines demarcate the antennal region. Refer to *Figure 2B* for a more detailed view. (**B and C**) *PKD1-1* (**B**) or *PKD2-1* (**C**) head gene expression revealed by in situ hybridization; anterior views. Compare cell position in B and C to panel A. White dashed lines demarcate the antennal region. (**D and E**) Immunostaining of 3xHA-Palmi-tdTomato expressed under the *PKD1-1* (**D**) or the *PKD2-1* (**E**) promoter; ventral views. Counterstaining against acetylated tubulin (AcTub) is in white. Green arrowheads in C and D point to CR cilia (green), MS1 (red) or MS2 cilia (yellow).

DOI: https://doi.org/10.7554/eLife.36262.017

The following source data and figure supplements are available for figure 3:

**Figure supplement 1.** *Platynereis* PKD1-1 and PKD2-1 belong to the TRPP/PKD1 superfamily.

DOI: https://doi.org/10.7554/eLife.36262.018

**Figure supplement 1—source data 1.** Source data of phylogenetic analysis, including alignment and tree files and nucleotide sequences of all *Platynereis* PKD genes analysed.

DOI: https://doi.org/10.7554/eLife.36262.019

**Figure supplement 2.** *PKD2-1* is expressed in CR and other penetrating ciliated neurons.

DOI: https://doi.org/10.7554/eLife.36262.020

**Figure supplement 3.** *PKD1-1* is expressed in CR and pygPB[unp] neurons.

DOI: https://doi.org/10.7554/eLife.36262.021

*1982*; *Purschke et al., 2017*; *Schlawny et al., 1991*; *Windoffer and Westheide, 1988*). In leeches, these cells (called 'S hairs') were suggested to be involved in the detection of water movement (*Friesen, 1981*). Morphologically, CR neurons are also similar to hair cells in vertebrates, which also can detect water movement (*Engelmann et al., 2000*). Further comparative work is needed to clarify the evolutionary relationships of these mechanosensory cell types (*Fritzsch et al., 2007*).

## A neuronal function of polycystins in *Platynereis* mechanosensation

At the molecular level, we uncovered an essential function of PKD1-1 and PKD2-1 in the transduction of hydrodynamic signals by CR neurons. The role of polycystins (for the phylogeny, see *Figure 3—figure supplement 1*) in neuronal mechanotransduction is poorly understood. Most studies focused on their functions in flow detection in endothelial cells in the kidney (*Nauli et al., 2008*; *Nauli et al., 2003*; *Pazour et al., 2002*; *Yoder et al., 2002*) and in nodal cilia during embryonic development (*Yoshiba et al., 2012*). In mammalian hair cells, polycystin-1 (PC1), a paralog of *Platynereis* PKD1-1,

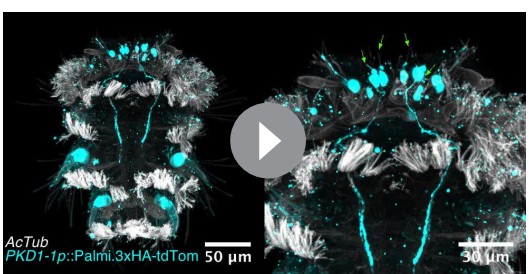

**Video 7.** Immunostained nectochaete larva against 3xHA-Palmi-tdTomato expressed under the *PKD1-1* promoter (*PKD1-1p*::Palmi-3xHA-tdTom) and counterstained against acetylated tubulin (AcTub). Right: full body view, Left: close up of the head region. Green arrows point to the hCR1 and hCR2 cilia.
DOI: https://doi.org/10.7554/eLife.36262.022

only plays a structural role and does not have a mechanosensory function (*Steigelman et al., 2011*). The normal morphology of the sensory ending of CR cells in *PKD1-1* and *PDK2-1* mutant animals suggests that PKD proteins are involved in the mechanotransduction cascade in *Platynereis* rather than having a structural role (*Figure 4—figure supplement 1*). In zebrafish, the polycystin PKD2L1 is involved in mediating mechanotransduction in a neuronal context. PKD2L1 is expressed in cerebrospinal-fluid-contacting neurons and is required for the detection of tail bending by these neurons and for normal tail movements (*Böhm et al., 2016*). Cerebrospinal-fluid-contacting neurons directly respond to mechanical stimulation and this sensitivity is dependent on PKD2L1 (*Sternberg et al., 2018*) . In *C. elegans*, *PKD2* and *LOV-1* (a *PKD1* paralog) are known to be required for neuronal mechanosensory function (*Barr and Sternberg, 1999*). Despite this requirement, calcium signalling is not impaired in mechanosensory neurons in the *C. elegans PKD2* mutant (*Zhang et al., 2018*). Future studies in the experimentally tractable *Platynereis* larvae, including calcium imaging in the mutants, could contribute to elucidating the role of these molecules in neuronal mechanotransduction.

## Startle responses in predator avoidance by planktonic larvae

The startle response in *Platynereis* larvae consists of the fast and synchronous activation of specific muscles and the arrest of all locomotor cilia. These responses were absent in larvae mutant for *PKD1-1* and *PKD2-1*. The use of *PKD2-1* mutant larvae allowed us to test the importance of the startle response in avoiding a natural predator, the copepod *Centropages typicus*. We attributed the large increase in predation on *PKD2-1* mutants by this copepod to a defective startle response. Mechanosensation is thought to be especially important for predator-prey interactions at small scales (*Andersen et al., 2016*; *Martens et al., 2015*). *Platynereis* larvae mutant for *PKD2-1* would not stop swimming even at close distance from the copepod, which uses mechanoreceptors in its antennae to localize prey (*Yen et al., 1992*). The lack of chaetal elevation would also make the mutants more susceptible to a copepod attack. Since *PKD2-1* is also expressed in other sensory cell types (e.g. PB neurons), we cannot rule out that other sensory defects contributed to the vulnerability of mutant larvae. Similar experiments with larvae mutant for *PKD1-1*, a gene specifically expressed in CR neurons, will be needed to strengthen our conclusion. The detailed analysis of hydrodynamic interactions between *Centropages* and *Platynereis* larvae, combined with the use of mutant *Platynereis* larvae represent an exciting future avenue of investigations.

## Potential circuit implementations of the graded and direction sensitive startle behaviour

Calcium imaging and the *PKD1-1* and *PKD2-1* mutants allowed us to link the hydrodynamic startle response to the CR neurons. By full-body electron microscopy we could then map the entire circuit downstream of CRs. The whole-body circuit reconstruction provided a framework to interpret the behaviour and propose hypotheses about how the different responses are controlled. We uncovered two major synaptic paths from CR neurons, one to ciliated effectors, and another one to a subset of somatic muscles. During the startle response, both ciliary and muscle effectors can be activated. Ciliary beating is controlled by a rhythmically activated ciliomotor system that drives alternating phases of ciliary activity and closures (*Tosches et al., 2014*; *Veraszstó et al., 2017*). We propose that slight disturbances in the surrounding medium activate the CR neurons in the head of the larva to drive the closure of all locomotor cilia in a coordinated fashion. This could be driven by a feed-forward circuit from head CRs to the cholinergic MC neuron and the activation of the cholinergic Loop and MN[ant] ciliomotor neurons via the CM interneurons. The beating of cilia generates a hydrodynamic

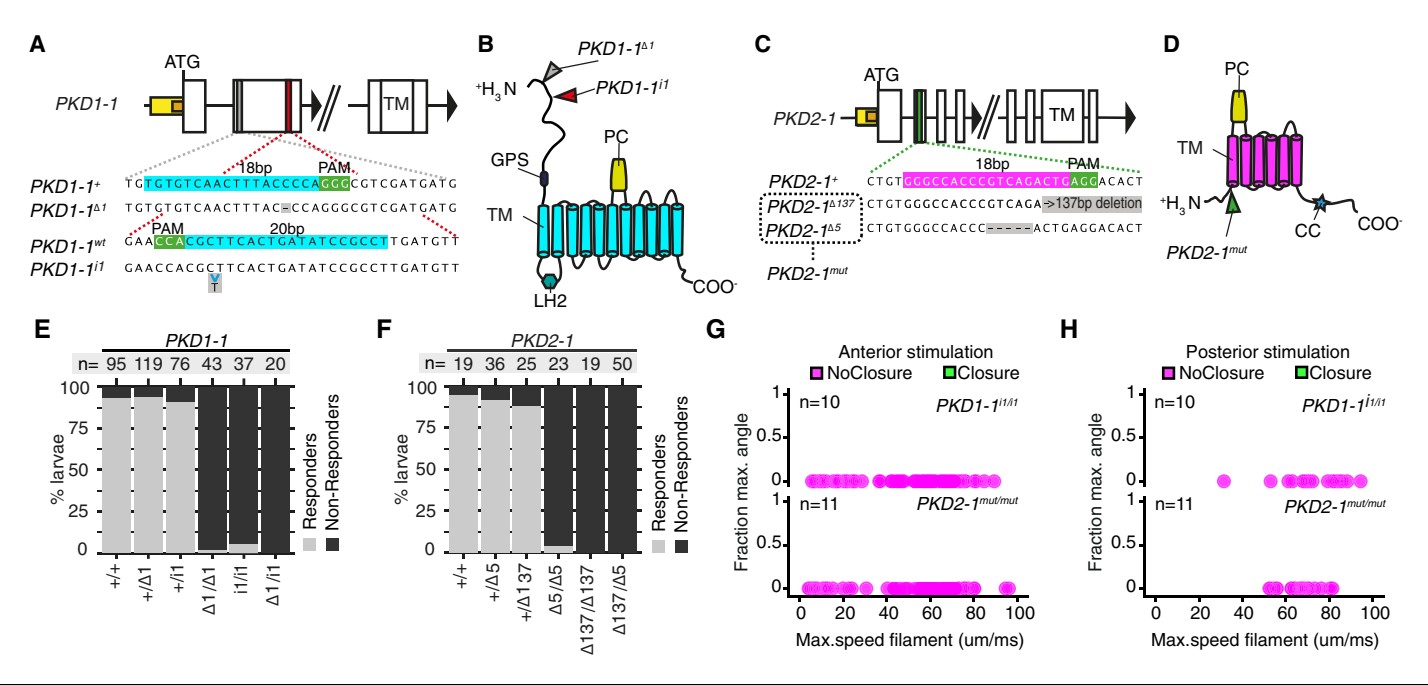

**Figure 4.** *PKD1-1* and *PKD2-1* are required for the startle response. (**A and C**) *PKD1-1* (**A**) and *PKD2-1* (**C**) genomic loci and close-up regions showing wildtype (+) sequences targeted with CRISPR/Cas9. The mutant alleles generated are also shown. White, yellow and orange boxes represent exons, promoters, and 5′UTRs, respectively. (**B and D**) PKD1-1 (**B**) or PKD2-1 (**D**) protein secondary structure. Main conserved domains are labelled. Mutation sites are indicated by arrowheads. (**E and F**) Stacked barplots of *PKD1-1* (**E**) or *PKD2-1* (**F**) mutant and wildtype (+/+) homozygote, heterozygote or trans-heterozygote larvae (%) startle response to touch stimulus. Data in (**E**) and in (**F**) from 18 or 14 batches, respectively. (**G and H**) Parapodial elevation angles of *PKD1-1$^{i1/i1}$* or *PKD2-1$^{mut/mut}$* larvae as a function of filament speed. Stimulation filament ca. 100 µm from anterior (**G**) or from posterior (**H**).

DOI: https://doi.org/10.7554/eLife.36262.023

The following source data and figure supplement are available for figure 4:

**Source data 1.** Phenotyping of PKD1-1 and PKD2-1 mutant larvae in the touch assay.
DOI: https://doi.org/10.7554/eLife.36262.025
**Source data 2.** Phenotyping of tethered PKD1-1 and PKD2-1 mutant larvae.
DOI: https://doi.org/10.7554/eLife.36262.026
**Figure supplement 1.** Cilia of hCR1 and hCR2 neurons in *PKD1-1$^{i1/i1}$* or *PKD2-1$^{mut/mut}$* larvae are morphologically similar to wildtype controls.
DOI: https://doi.org/10.7554/eLife.36262.024

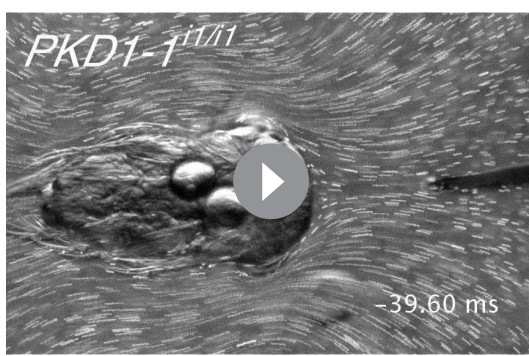

**Video 8.** No startle response in a PKD1-1$^{i1/i1}$ mutant l *Platynereis* larva exposed to a high-intensity stimulus. Time is relative to stimulus start.

DOI: https://doi.org/10.7554/eLife.36262.027

signal that can be detected by predators at a distance (*Kiørboe and Visser, 1999*). By shutting off cilia upon vibrations elicited by an approaching predator, the larva could avoid being detected (*Figure 9A*). The ability to cease ciliary beating upon mechanical stimulation has been reported in other ciliated larvae (*Mackie et al., 1976*) and could be a widespread predator avoidance mechanism.

At higher stimulation intensities, we observed the rapid closure of the ciliary bands and the simultaneous wide elevation of all parapodia. This defensive behaviour could be triggered by an attacking predator (*Figure 9C* and *Video 9*). The wiring diagram suggests that Rope neurons could simultaneously activate ciliomotor and trunk musclemotor neurons across all segments

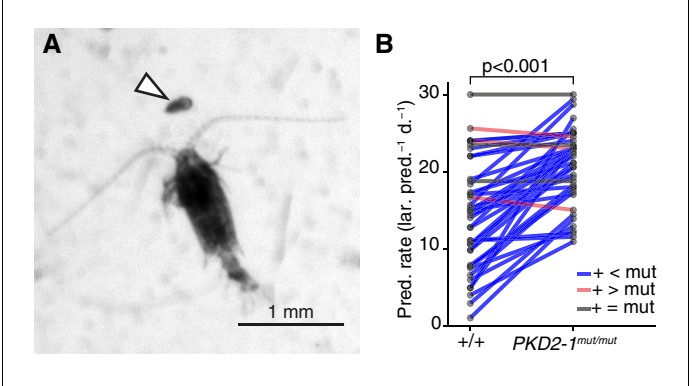

**Figure 5.** *PKD2-1* is required for predator avoidance and escape. (**A**) Adult *Centropages typicus* female approaching a nectochaete *Platynereis* larva (arrowhead). (**B**) Predation on wildtype (+/+) and *PKD2-1^{mut/mut}* larvae by *C. typicus*. Paired values are joined by blue, grey or red lines if predation rates were higher, equal or lower in mutant than in wildtype larvae, respectively; data from 42 trials with 12 batches. One-sided exact Wilcoxon-Pratt signed rank test, P = 5.2e-10.

DOI: https://doi.org/10.7554/eLife.36262.028

The following source data is available for figure 5:

**Source data 1.** Source data of predation experiments.

DOI: https://doi.org/10.7554/eLife.36262.029

upon strong stimulation of head CRs. This could lead to the observed coordinated contraction of muscles and ciliary closures (*Figure 1J* and *Figure 1—figure supplement 1*). Thus, based on their anatomy and connectivity, Rope neurons are candidate command interneurons in *Platynereis* (*Wiersma and Ikeda, 1964*). Their circuitry resembles that of giant fibre systems in a variety of animals, such as the GF system in *Drosophila*, the lateral and medial giants in crayfish or the M cells in vertebrates (reviewed in *Hale et al., 2016*). Adult *Platynereis* worms also have a giant fibre system involved in a startle reflex (*Smith, 1957*) but how these relate to the Rope neurons is unclear.

Beside the Rope neurons, Split interneurons constitute a second pathway for synchronous and intersegmental muscle contraction (*Figure 6B*, *Figure 7A*, and *Figure 8C*). Like Rope neurons, Split neurons receive input from head CRs and target musclemotor neurons in all segments (*Figure 7A* and *Figure 6—figure supplement 2*). These two systems could function in parallel to produce the complete startle response (*Figure 9C*). A similar mechanism has been suggested to be at work during whole-body shortening in the leech. Here, the fast-conducting S-cells trigger the initial muscle contraction followed by the activation of a slower pathway mediating extensive body shortening (*Kristan et al., 2005*; *Shaw and Kristan, 1995*).

The Rope and Split pathways may also implement different aspects of a startle behaviour depending on context or the nature, direction or intensity of the stimulus (*Fotowat et al., 2009*; *Kramer et al., 1981*; *Liu and Fetcho, 1999*)(*Figure 8*). For example, Split but not Rope neurons are also targeted by posterior CRs. This pathway may activate muscles independent of ciliary closures, as observed upon posterior stimulation. Such a response may help the larva to deter and escape a predator approaching from the

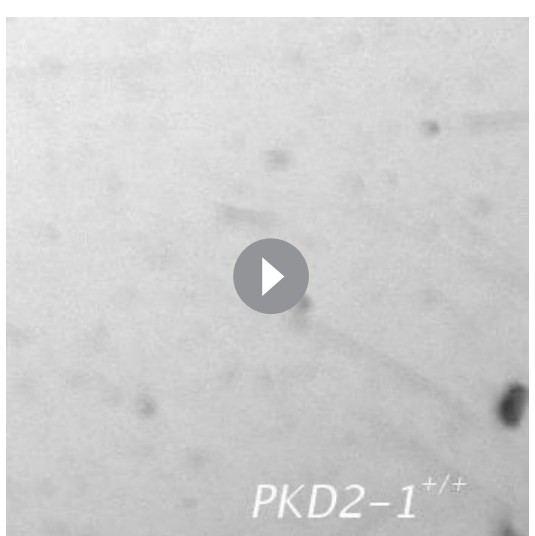

**Video 9.** Copepod attack on either a wild type or a *PKD2-1^{mut/mut}* mutant *Platynereis* larva.

DOI: https://doi.org/10.7554/eLife.36262.030

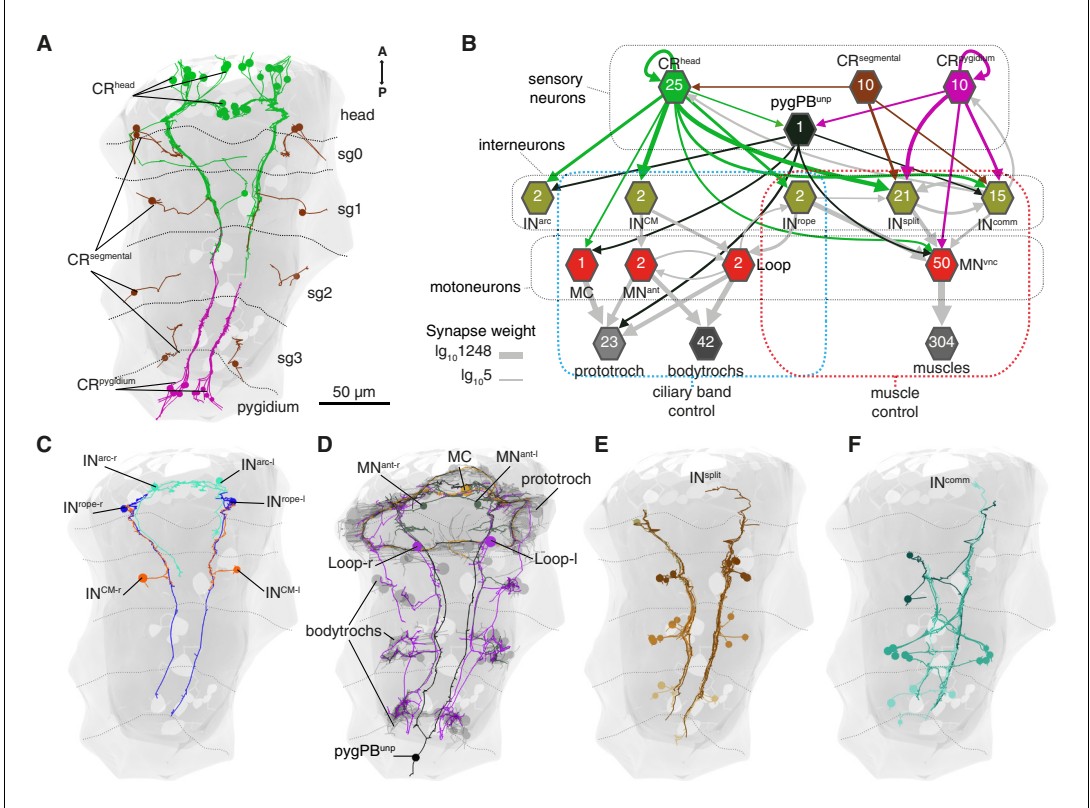

**Figure 6.** Wiring diagram of CR neurons. (**A**) All CRs in the nectochaete larva reconstructed from an EM volume. Head, segmental and pygidial groups are differently coloured. Segment (sg) boundaries are indicated by dotted lines. (**B**) Sensory-motor network from CR neurons to muscles and locomotory cilia. Hexagons and arrows represent neuron groups and their synaptic connections, respectively. Numbers inside hexagons indicate the number of neurons grouped in each node. Interactions with less than five synapses were filtered out in this display. Arrow line thickness (synapse weight) is equal to the common logarithm of the number of synapses (scale is shown on the bottom left of the panel). (**C**) EM reconstruction of $IN^{rope}$, $IN^{arc}$ and $IN^{CM}$ neurons. (**D**) Ciliomotor neurons targeted by CRs or by $IN^{CM}$ (**E**) Ipsilateral split neurons ($IN^{split}$) (**F**) Commissural interneurons ($IN^{comm}$). Ventral view in A, C-F. Segment boundaries as defined in A are overlaid in C-F.

DOI: https://doi.org/10.7554/eLife.36262.031

The following source data and figure supplements are available for figure 6:

**Source data 1.** Full connectivity matrix, grouped.
DOI: https://doi.org/10.7554/eLife.36262.035

**Source data 2.** Full connectivity matrix, non-grouped.
DOI: https://doi.org/10.7554/eLife.36262.036

**Figure supplement 1.** CR neurons target ipsilateral and commissural interneurons innervating VNC motoneurons.
DOI: https://doi.org/10.7554/eLife.36262.032

**Figure supplement 2.** VNC musclemotor neuron classes in the CR neural network.
DOI: https://doi.org/10.7554/eLife.36262.033

**Figure supplement 3.** Synaptic connections in the startle circuit.
DOI: https://doi.org/10.7554/eLife.36262.034

posterior end (*Figure 9D*). Differences in connectivity of anterior and posterior sensory neurons to the downstream circuits have also been observed in *C. elegans* (*Chalfie et al., 1985*) and crayfish (*Wine and Krasne, 1972*). The study of the startle response in different contexts (swimming vs crawling states) combined with neuron imaging and functional interference will be needed to elucidate the contributions of these different pathways.

## Bilateral coordination of the startle response

A hallmark of the startle response in *Platynereis* is its bilateral symmetry, both in timing and in the extent of muscle contraction (*Figure 1* and *Figure 1—figure supplement 1*). The proximity of the

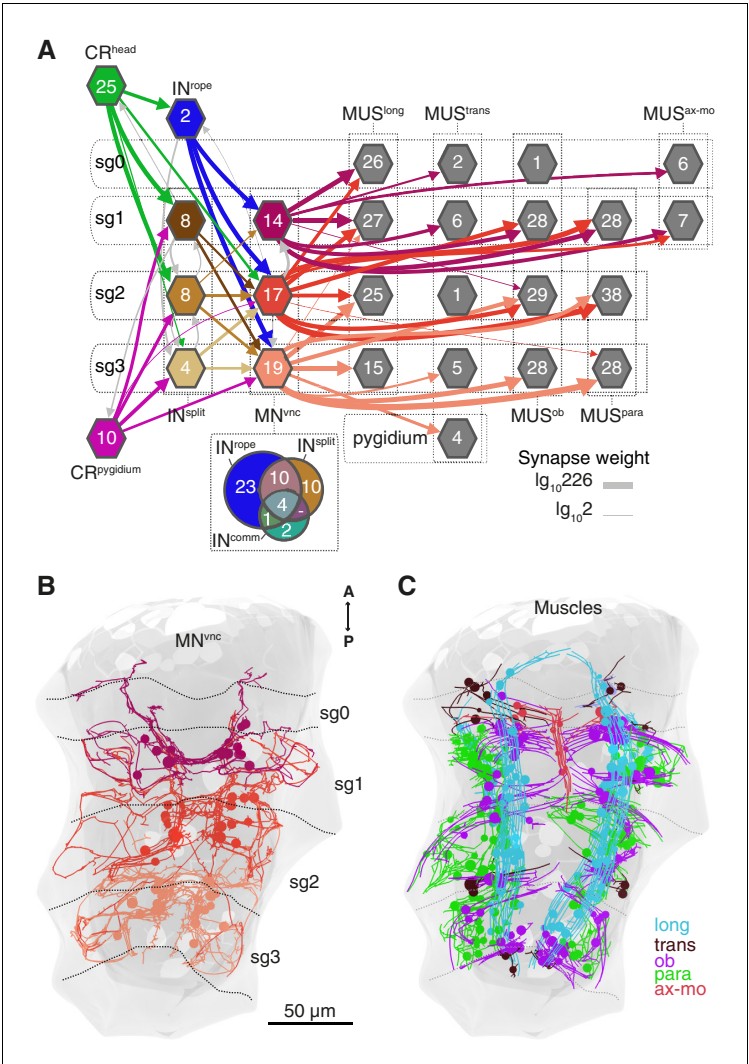

**Figure 7.** Segmental coordination motifs and musclemotor neuron diversity in the startle circuit. (**A**) Muscle network downstream of IN[rope] and IN[split] sorted by segmental location and muscle type. Arrows represent synaptic connections and are coloured based on the source neuronal group (represented by hexagons). Venn diagram at the bottom of the network shows the number of common or unique VNC motoneurons (MN[vnc]) targets of IN[rope], IN[split], and IN[comm]. (**B**) MN[vnc] targeted by Rope and Split interneurons coloured by segmental location. (**C**) Muscles targeted by MN[vnc] neurons coloured by type (long, longitudinal; trans, transvers; ob, oblique; para, parapodial; ax-mo, axochord-mouth). Segment (sg) boundaries are indicated by dotted lines in A and C. Arrow line thickness (synapse weight) in A is equal to the common logarithm of the number of synapses.
DOI: https://doi.org/10.7554/eLife.36262.038

sensory cilia of all head and pygidial CRs suggests that both the left and right members are stimulated in most cases, as observed during calcium imaging of hCR1 cells (*Figure 2E–G*). Since the circuitry downstream of CR neurons is bilaterally symmetric (*Figure 8*), a coincident activation of left and right CRs would result in a synchronous response. But even if asymmetries in CR activation occur, circuit motifs in the network could ensure the left-right coordination of muscle and ciliary effectors. Ciliary coordination may partially rely on CM interneurons targeting ciliomotor MN[ant] cells of both body sides (*Figure 8*). Coincident left-right muscle contraction could be ensured by the connection of a single Rope neuron to both ipsi- and contralateral decussating motoneurons. Split neurons have a similar bilateral connectivity pattern (*Figure 8*). A similar network motif is present in the *Drosophila* larva, where a single commissural interneuron innervates motoneurons on both body sides (*Fushiki et al., 2016*). The commissural interneurons (IN[comm]) may also contribute to left-right

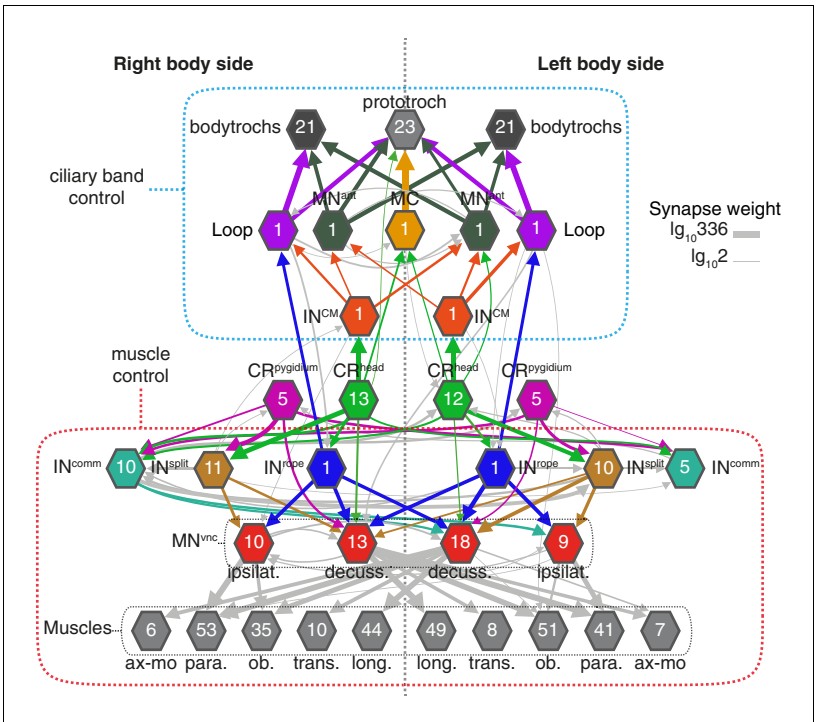

**Figure 8.** Bilateral coordination motifs in the startle circuit. Muscle and cilia network downstream of CR^head and CR^pygidium sorted by body side. Numbers inside hexagons indicate the number of neurons grouped in each node. Interactions with less than two synapses were filtered out. Arrow line thickness (synapse weight) is equal to the common logarithm of the number of synapses.

DOI: https://doi.org/10.7554/eLife.36262.039

coordination. In *Drosophila*, the amplitude of muscle contractions on both body sides is controlled by ascending commissural neurons (*Heckscher et al., 2015*).

The study of the *Platynereis* larva allowed us to analyse in a whole-body context the molecular, cellular and circuit components of an ecologically relevant behaviour. We uncovered several motifs in this circuit that show similarities to startle circuits in other animals. The evolutionary relationship of the *Platynereis* circuit to similar circuits in

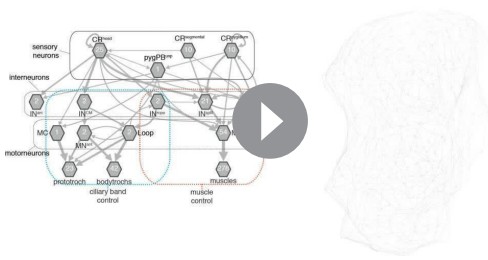

**Video 10.** Animation of the wiring diagram and neuronal morphology reconstructed from serial electron microscopy data. Colours in wiring diagram (left) correspond to colours of neurons in 3D model of the larva (right).

DOI: https://doi.org/10.7554/eLife.36262.040

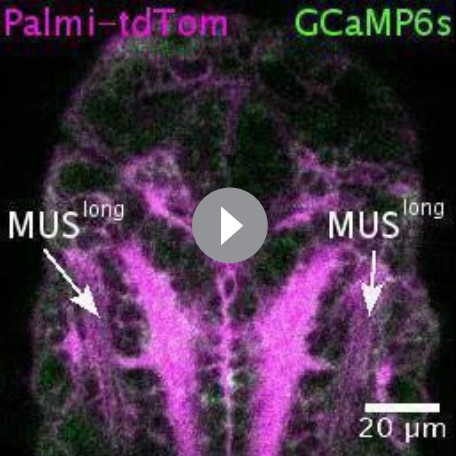

**Video 11.** Calcium imaging from larval muscles during a startle response. The larva is ubiquitously expressing Palmi-tdTomato and GCaMP6s. The contractions of the longitudinal, oblique and parapodial muscles are visible.

DOI: https://doi.org/10.7554/eLife.36262.041

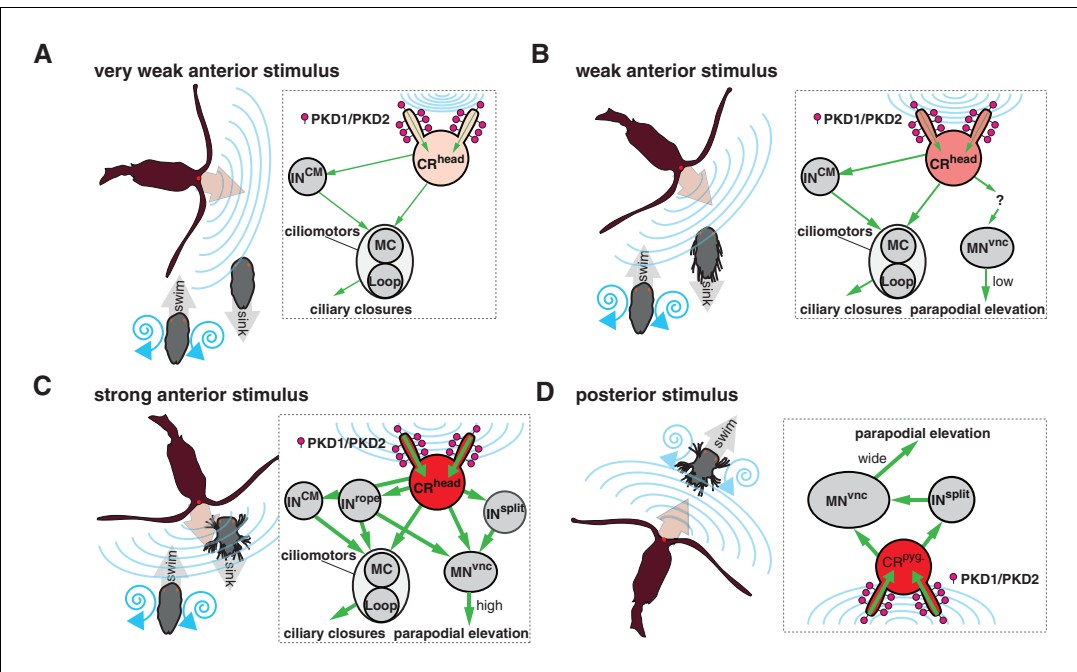

**Figure 9.** Summary schematics of the proposed neural implementations behind the startle response in different predator-prey scenarios. (A–D) Different interaction scenarios between a predatory copepod (brown) and a nectochaete *Platynereis* larva (grey) are schematized on the left side of each panel. Blue lines schematize hydrodynamic signals produced by the copepod or the larva. Arrows indicate the direction of movement. Inset on the right of each panel shows the proposed cellular/circuit implementation behind the corresponding larval response shown to the left. Green arrows indicate the direction of propagation of the signal. PKD1-1 and PKD2-1 (symbolized as lollipops) localize to CRs, which transduce a signal of different intensity (shown with different tones of red) depending on the strength of the hydrodynamic signal (blue lines).

DOI: https://doi.org/10.7554/eLife.36262.042

other bilaterians is currently unclear. Comparative connectomics of whole-body circuits has great potential to uncover circuit homologies. A recent analysis of the *Ciona intestinalis* larval connectome identified a putative startle circuit based on anatomy and connectivity alone that was suggested to be homologous to the Mauthner circuit of vertebrates (*Ryan et al., 2017*; *Ryan et al., 2016*). To understand circuit evolution, such approaches will need to be complemented with genetic and behavioural experiments and the analysis of the constituent cell types. Only comparisons at these different levels and at varying phylogenetic distances will allow us to reconstruct the evolution of circuits.

# Materials and methods

## Key resources table

| Reagent type (species) or resource | Designation | Source or reference | Identifiers | Additional information |
|---|---|---|---|---|
| Gene (P*latynereis dumerilii*) | *PKD1-1* | This paper | GenBank_Acc#: MH035679 | |
| Gene (*P. dumerilii*) | *PKD2-1* | This paper | GenBank_Acc#: MH035677 | |
| Gene (*P. dumerilii*) | *PKD2-2* | This paper | GenBank_Acc#: MH035678 | |
| Gene (*P. dumerilii*) | *LOV-1* | This paper | GenBank_Acc#: MH035680 | |

*Continued on next page*

*Continued*

| Reagent type (species) or resource | Designation | Source or reference | Identifiers | Additional information |
|---|---|---|---|---|
| Strain, strain background (*P. dumerilii*) | Wild Type | Tübingen cultures, Max Planck Institute for Developmental Biology | NCBITaxon:6359 | |
| Strain, strain background (*P. dumerilii*) | *PKD1-1*$^{\Delta1/\Delta1}$ | This paper | | Progenitor:Tübingen WT strain using *PKD1-1*_truT1 gRNA |
| Strain, strain background (*P. dumerilii*) | *PKD1-1*$^{i1/i1}$ | This paper | | Progenitor:Tübingen WT strain using *PKD1-1*_T5 gRNA |
| Strain, strain background (*P. dumerilii*) | *PKD2-1*$^{\Delta137/\Delta137}$; *PKD2-1*$^{mut/mut}$ | This paper | | Progenitor:Tübingen WT strain using *PKD2-1*_truT2 gRNA |
| Strain, strain background (*P. dumerilii*) | *PKD2-1*$^{\Delta5/\Delta5}$; *PKD2-1*$^{mut/mut}$ | This paper | | Progenitor:Tübingen WT strain using *PKD2-1*_truT2 gRNA |
| Strain, strain background (*P. dumerilii*) | *PKD2-1*$^{\Delta137/\Delta5}$; *PKD2-1*$^{mut/mut}$ | This paper | | Progenitor:*PKD2-1*$^{\Delta5}$ and *PKD2-1*$^{\Delta137}$ allele-carrying strains. |
| Strain, strain background (*Centropages typicus*) | Wild Type | National Institute for Aquatic Resources culture, Technical University of Denmark | NCBITaxon: 463189 | |
| Antibody | Goat anti-Mouse IgG (H + L) Cross-Adsorbed Secondary Antibody, Alexa Fluor 488 | ThermoFisher Scientific | Cat#:A11001 | (1:250) |
| Antibody | Goat anti-Rabbit IgG (H + L) Cross-Adsorbed Secondary Antibody, Alexa Fluor 633 | ThermoFisher Scientific | Cat#:A21070 | (1:250) |
| Antibody | Monoclonal Anti-Tubulin, Acetylated antibody produced in mouse | Sigma-Aldrich | Cat#:T6793, RRID:AB_477585 | (1:250) |
| Antibody | HA-Tag (C29F4) Rabbit mAb | Cell Signaling Technology | Cat#:3724P | (1:250) |
| Recombinant DNA reagent | pUC57-*PKD2-1p*::Palmi-3xHA-tdTomato (plasmid) | This paper | | Promoter construct: injected at 250 ng/µl |
| Recombinant DNA reagent | pUC57-*PKD1-1p*::Palmi-3xHA-tdTomato (plasmid) | This paper | | Promoter construct: injected at 250 ng/µl |
| Recombinant DNA reagent | pUC57-T7-RPP2-tdTomato-P2A-GCaMP6 (plasmid) | This paper | | Used for generating tdTomato-P2A-GCaMP6s mRNA |
| Recombinant DNA reagent | pMJ920 (plasmid) | DOI: 10.7554/eLife.00471 | Addgene_plasmid: #42234 | Laboratory of Jennifer Doudna |
| Recombinant DNA reagent | pUC57-T7-RPP2-hSpCas9-HA-2XNLS-GFP (plasmid) | This paper | | Used for generating Cas9-GFP mRNA |
| Recombinant DNA reagent | pGP-CMV-GCaMP6s (plasmid) | DOI: 10.1038/nature12354 | Addgene_plasmid: #40753 | Laboratory of Douglas Kim |
| Recombinant DNA reagent | pX335-U6-Chimeric_BB-CBh-hSpCas9n(D10A) (plasmid) | DOI: 10.1126/science.1231143 | Addgene_plasmid: #42335 | Laboratory of Feng Zhang |
| Recombinant DNA reagent | DR274 (plasmid) | DOI: 10.1038/nbt.2501 | Addgene_plasmid: #42250 | Laboratory of Keith Joung |
| Recombinant DNA reagent | *PKD2-1*_truT2 (plasmid) | This paper | | Derived from Addgene plasmid 42250.Used for gRNA synthesis |

*Continued on next page*

*Continued*

| Reagent type (species) or resource | Designation | Source or reference | Identifiers | Additional information |
|---|---|---|---|---|
| Recombinant DNA reagent | *PKD1-1*_truT1 (plasmid) | This paper | | Derived from Addgene plasmid 42335.Used for gRNA synthesis |
| Recombinant DNA reagent | *PKD1-1*_T5 (plasmid) | This paper | | Derived from Addgene plasmid 42335.Used for gRNA synthesis |
| Recombinant DNA reagent | pCRII-*PKD2-1* (plasmid) | This paper | | Used for synthesizing WMISH probes |
| Recombinant DNA reagent | pCRII-*PKD1-1* (plasmid) | This paper | | Used for synthesizing WMISH probes |
| Sequence-based reagent | *PKD1-1*-WMISH forward primer (5'- > 3') | This paper | | TGTCTTTGTAACTGTTGGTCTGGT |
| Sequence-based reagent | *PKD1-1*-WMISH reverse primer (5'- > 3') | This paper | | ATGTTCCTCAGAGATTCCTTCATC |
| Sequence-based reagent | *PKD2-1* WMISH forward primer (5'- > 3') | This paper | | TCCTCGTCATCATAATATCCATTG |
| Sequence-based reagent | *PKD2-1* WMISH reverse primer (5'- > 3') | This paper | | CTCTCTTTGTTGAGTTGGTCCTTT |
| Sequence-based reagent | *PKD1-1* T5 (gRNA) | This paper | | 20 ng/µl; AGGCGGATATCAGTGAAGCG; It generated i1 allele in *PKD1-1*. |
| Sequence-based reagent | *PKD1-1* truT1 (gRNA) | This paper | | 20 ng/µl; TGTGTCAACTTTACCCCA. It generated Δ1 allele in *PKD1-1*. |
| Sequence-based reagent | *PKD2-1* truT2 (gRNA) | This paper | | 20 ng/µl; GGGCCACCCGTCAGACTG. It generated Δ137 and Δ5 alleles in *PKD2-1*. |
| Sequence-based reagent | *PKD*1-1_Genotyping-truT1andT5-forward primer (5'- > 3') | This paper | | TCAAACTGGTCAAGATTAAATTCCAGA |
| Sequence-based reagent | *PKD*1-1_Genotyping-truT1andT5-reverse primer (5'- > 3') | This paper | | TCTATTTCACTAATGTTGTTCCTGATG |
| Sequence-based reagent | *PKD*1-1 truT1-sequencing primer (5'- > 3') | This paper | | TAAGTGAAGGTCACATACTCGTCAGT |
| Sequence-based reagent | *PKD*1-1 T5-sequencing primer (5'- > 3') | This paper | | TATGAGACTGAATGCACAATAGAGTTT |
| Sequence-based reagent | *PKD*2-1_Genotyping-truT2-forward primer (5'- > 3') | This paper | | CCCTTTGTGAGCAGGAGATGCCCTGC |
| Sequence-based reagent | *PKD*2-1_Genotyping-truT2-reverse primer (5'- > 3') | This paper | | CATGACCTGAGTGTAGTAGTACATGGT |
| Sequence-based reagent | *PKD*2-1 truT2-sequencing primer (5'- > 3') | This paper | | AACCTTCAAATATGTTCACTACAATCC |
| Sequence-based reagent | BamHI-PKD1-1promoter forward primer (5'- > 3') | This paper | | ATGGATCCGGAAGCCTGATAACAAAGTGAGTAGGAA |
| Sequence-based reagent | AscI-PKD1-1promoter reverse primer (5'- > 3') | This paper | | AAGGCGCGCCGCTCCTCCTCCAAGTGGCTCTACCATCTCCGTCTGTATC |
| Sequence-based reagent | BamHI-PKD1-1promoter forward primer (5'- > 3') | This paper | | ATGGATCCAATAAAATTTGAACCGAGGCCAATGGA |
| Sequence-based reagent | AscI-PKD1-1promoter reverse primer (5'- > 3') | This paper | | AAGGCGCGCCGCTCCTCCTCCGGGGCGTGACATCAGCCAGTAGTTGGT |
| Commercial assay or kit | QuickExtract | Epicentre,US | Cat#:QE09050 | |

*Continued on next page*

*Continued*

| Reagent type (species) or resource | Designation | Source or reference | Identifiers | Additional information |
|---|---|---|---|---|
| Commercial assay or kit | MEGAshortscript T7 Transcription Kit | Ambion, ThermoFisher Scientific | Cat#:AM1354 | |
| Commercial assay or kit | mMESSAGE mMACHINE T7 ULTRA Transcription Kit | Ambion, ThermoFisher Scientific | Cat#:AM1345 | |
| Commercial assay or kit | MEGAclear Transcription Clean-Up Kit | Ambion, ThermoFisher Scientific | Cat#:AM1908 | |
| Chemical compound, drug | Mecamylamine | Sigma-Aldrich | Cat#:M9020 | 500 µM |
| Software, algorithm | Fiji | NIH | RRID:SCR_002285 | |
| Software, algorithm | R Project for Statistical Computing | R Foundation | RRID:SCR_001905 | |
| Software, algorithm | Imaris Version 8.0.0 | Bitplane, UK. | RRID:SCR_007370 | |
| Software, algorithm | CATMAID | DOI: 10.1093/ bioinformatics/btp266 | RRID:SCR_006278 | |
| Software, algorithm | PhyML | DOI: 10.1093/sysbio/ syq010 | RRID:SCR_014629 | |
| Software, algorithm | Gblocks | DOI: 10.1080/ 10635150701472164 | RRID:SCR_015945 | |
| Other | Wormglu | GluStitch Inc. | | |
| Other | Fluoresbrite Multifluorescent Microspheres 1.00 µm | Polysciences Inc, US | Cat#:24062–5 | (1:1000) |

## Animal culture and behavioural experiments

*Platynereis dumerilii* wild type (Tübingen, NCBITaxon:6359) strain and the mutant lines derived from it were cultured in the laboratory as previously described (*Fischer and Dorresteijn, 2004*). The nectochaete larval stage (approximately 72 hr post fertilization) was utilized for all experiments and raised at 18°C on a 16:8 hr light/dark cycle in glass beakers. Filtered natural sea water (fNSW) was used for all the experiments.

Adult stages of the marine planktonic copepod *Centropages typicus* were used as rheotactic predators for wild type and *PKD2-1* mutant larvae. Copepods were supplied from a continuous culture at the National Institute for Aquatic Resources (Technical University of Denmark, DTU). Specimens of *C. typicus* were originally isolated from zooplankton samples collected in the Gullmar fjord (Sweden) by vertical tows with plankton nets (500 µm mesh). Cultures of *C. typicus* were kept in 30 L plastic tanks with sterile-filtered seawater (FSW, salinity 32 ppt), gently aerated, at 16 ± 1°C in dark. Copepods were fed *ad libitum* with a mix of phytoplankton (the cryptophyte *Rhodomonas sp.*, the diatom *Thalassiosira weissflogii* and the autotrophic dinoflagellates *Heterocapsa triquetra*, *Prorocentrum minimum* and *Gymnodinium sanguineum*) and with the heterotrophic dinoflagellate *Oxyrrhis marina*. Phytoplankton cultures were kept in exponential growth in B1 culture medium and maintained at 18°C and on a 12:12 hr light/dark cycle in glass flasks. *O. marina* was fed the cryptophyte *Rhodomonas salina* and maintained at 18°C in 2 L glass bottles. Behavioural experiments were carried out at room temperature unless otherwise indicated.

## Kinematics of startle behaviour

Larvae were relaxed in 50–100 mM $MgCl_2$ 10 min before tethering them to a Ø3.5 cm glass-bottom dish (HBST-3522, Willco Wells) with a non-toxic glue developed for *C. elegans* (Wormglu, GluStitch Inc). Care was taken to minimize or to avoid gluing ciliary bands, sensory cilia, parapodia, head and pygidium. Prior to the experiment, the glued larvae were assessed for the startle response with a gentle vibration to verify that relevant structures were unhindered, and the animal was healthy and in a swimming mode. 1 µm multi-fluorescent beads (24062–5, Polysciences) were diluted in 5% BSA to 1:100, sonicated for 1 min and added to the glued larva preparation at a 1:10 dilution. The

experiments were done in a final volume of 2.5 ml. Recordings were done with an AxioZoom V.16 (Carl Zeiss GmbH, Jena) and responses were recorded with a digital CMOS camera ORCA-Flash-4.0 V2 (Hamamatsu). An HXP 200 fluorescence lamp at maximum level (using the Zeiss 45 mCherry filter) was switched on only during each recording (lasting max. 7 s each) to visualize the beads.

To generate water-borne vibrations, a thin 3–5 cm tungsten needle (RS-6063, Roboz) was glued to a shaft-less vibration motor (EXP-R25-390, Pololu), which was switched on for a defined time interval (1–35 ms) and induced the needle to vibrate. The motor was switched on and off with a custom script via an Arduino microcontroller (Arduino UNO R3, Arduino). The probe was positioned in focus at a defined distance from the larva with the manual micromanipulator US-3F (Narishige, Japan). A defined set of stimulation values were used for each of the animals tested, the order of the values was randomized for each larva. Between each stimulation attempt, the larva was left to rest for 1 min. Recordings were discarded if the larva was not in swimming mode while being stimulated. Stimulus start was defined from the videos by the onset of probe movement. The larvae were still alive and visibly healthy one day after being glued. All wild type larvae tested came from different batches. In the case of mutants, 10 larvae from two $PKD1$-$1^{i1/i1}$ batches and 11 larvae from three $PKD2$-$1^{mut/mut}$ batches were tested. Each tested larva was genotyped after the experiment.

The parapodial elevation angle was measured as the movement of the distal end of one of the parapodia in the first segment relative to its base that occurred from the rest position prior to stimulation to the maximal elevation achieved upon stimulation. The value was normalized to the maximum angle recorded for a given larva. Filament speed was calculated as the maximum displacement observed between consecutive frames normalized to the recording speed. Onset of ciliary band closures was defined as the moment after stimulation the flow of fluorescent beads stopped in the vicinity of the cilia. For bodytroch closures this was assessed from the posterior-most band (telotroch). Onset of parapodial elevation was defined as the moment after stimulation when the first segment parapodium (from which the elevation angle was measured) started its angular elevation. The slow elevation speed for LowE responses in many cases made it difficult to assess with precision the start of the elevation movement. The data were analysed with custom R scripts (*Bezares-Calderón, 2018*; copy archived at https://github.com/elifesciences-publications/Bezares_et_al_2018).

## Startle assay in freely swimming animals

4 ml of phototactic nectochaete larvae were transferred to a Ø5 cm glass-bottom dish (GWSB-5040, Willco Wells) equipped with a vibrating shaft-less motor glued to its bottom. The motor was activated for 100 ms via an Arduino microcontroller with a custom-written script. The same microscopy equipment was used as for the kinematics experiments, but the recording speed was set to 15 fps. A long-pass filter was placed between the light source and the dish to minimize phototaxis. The speed of startled larvae was calculated as the Pitagorean distance over time unit, and the area was measured from the thresholded shape of the larva.

## Whole mount in situ hybridization and immunochemistry

In situ hybridization probes for detecting *PKD1-1* and *PKD2-1* were synthesized from 1 Kb gene fragments subcloned into pCRII vectors (K206001, ThermoFisher) (*Figure 1—figure supplement 1—source data 1*), or from plasmids from an in-house EST library. Whole mount in situ hybridization was done as previously described (*Conzelmann et al., 2011*). The *PKD1-1* and *PKD2-1* promoters (fragment sizes: 2.5 Kb and 1.5 Kb, respectively) were amplified and cloned upstream of *3xHA-Palmi-tdTomato*. Larvae injected with promoter constructs (ca. 250 ng/µl) were analysed for reporter expression at 3 days post fertilization using an AxioImager Z.1 fluorescence wide-field microscope (Carl Zeiss GmbH, Jena) and immediately fixed for immunostainings. The protocol followed for immunostaining of HA-tagged reporters was recently described (*Verasztó et al., 2017*). Specimens were imaged with a LSM 780 NLO Confocal Microscope (Zeiss, Jena).

## Generation of *PKD1-1* and *PKD2-1* mutants with CRISPR/Cas9

The SpCas9-GFP ORF was kindly provided by Jennifer Doudna (Addgene plasmid #42234) and was subcloned into a custom-made plasmid construct tailored for enhanced mRNA expression in *Platynereis*. sgRNAs were designed using ZiFiT (*Sander et al., 2007*) and cloned into either the DR274 plasmid (kindly provided by Keith Joung, Addgene plasmid #42250) or pX335-U6-Chimeric-BB-CBh-

hSpCas9n (D10A) (kindly provided by Feng Zhang, Addgene plasmid #42335). sgRNAs were synthesized from PCR templates (*Figure 3—figure supplement 1—source data 1*) with the MEGAshortscript T7 kit (AM1354, Ambion) and purified using the MEGAclear Transcription Clean-up kit (AM1908, Ambion). Zygotes were injected with a mix of 300 ng/μl SpCas9-GFP mRNA and 20 ng/μl sgRNA dissolved in DNAse/RNase-free water (10977–049, ThermoFisher). 1-day-old larvae were screened for green fluorescence and gDNA was extracted from single larvae in 4 μl QuickExtract (QE) Solution (QE09050, Epicentre). Mutations were detected by PCR and Sanger sequencing. Adult (atokous) worms injected with effective sgRNAs were genotyped by clipping the pygidium following a previously published protocol (*Bannister et al., 2014*), but using 20 μl of QE solution for gDNA extraction. Worms carrying desired mutations were outcrossed to the wild type strain for the first two generations before using them for experiments.

### Phenotyping of mutants
Phototactic nectochaete larvae were transferred to a Petri dish and a single randomly selected larva was placed on a slide in 40 μl fNSW. The larva was scored for the startle response by touching it on the head with a tungsten needle. Only swimming larvae were assayed. For easier interpretation, phenotyped larvae were classified into responders or non-responders. Single larvae were genotyped as described in the preceding section and only after assignation to a phenotype group. The *PKD2-1$^{D137}$* allele was detectable by gel electrophoresis, and thus only determined by sequencing in ambiguous cases.

### Calcium imaging
The *Palmi-3xHA-tdTomato-P2A-GCaMP6s* construct was assembled by restriction cloning from a plasmid kindly provided by Douglas Kim (*GCaMP6s*, Addgene plasmid #40753) and Martin Bayer (*tdTomato*). *mRNA* was synthesized with the mMESSAGE mMACHINE T7 Transcription Kit (AM1344, Ambion). Animals injected with 2 μg/μl mRNA dissolved in RNAse-free water were tethered as described for the startle response quantification experiments but using in this case a Ø5 cm glassbottom dish in 5 ml volume. Tethered larvae were stimulated from the anterior with the filament set at a fixed value that invariably triggered the response. However, the maximum filament speed could not be directly measured for these experiments. Stimulus start was determined from the bright field channel (*Video 4*). In order to minimize muscle contractions, 500 μM Mecamylamine (M9020, Sigma) was added to the dish with the tethered larva. For calcium imaging of muscles no anaesthetic compound was used. Slight X-Y shifts were corrected using descriptor-based series registration (*Preibisch et al., 2010*). Shifts in Z were accounted for with the tdTomato signal using the following formula:

$$\Delta R/R = \frac{F(t)_{GCaMP6s} x F0_{tdTom}}{F0_{GCaMP6s} x F(t)_{tdTom}}$$

taken from (*Böhm et al., 2016*), where F0 is the average fluorescence prior to stimulation calculated from 1/2 the length of the pre-stimulation recorded period. Background signal (non-tissue signal) was used to adjust F0 and F(t). Most recordings were done at 4 Hz, 3 recordings at 2.6 Hz.

### Circuit reconstruction
CR neurons and the downstream circuit were reconstructed from a ssTEM stack previously reported (*Randel et al., 2015*) using the collaborative annotation tool CATMAID (RRID:SCR_006278; *Saalfeld et al., 2009*; *Schneider-Mizell et al., 2016*). Sensory endings in the left pygidial cirrus could not be imaged and thus CR neurons were not identified there. Chemical synapses were defined as a discrete accumulation of vesicles in the inner side of the presynaptic membrane (*Figure 6—figure supplement 3*). The volume of each synapse was not considered (i.e. each synapse had a weight of 1, independently of how many sections it spanned), and thus the resulting network is a conservative representation of the synaptic strength between neurons. Any given neuron downstream of the CR neurons was included in the analysis if it had three or more synaptic contacts from at least 2 CR neurons. After applying this filter, additional neurons with only 1 or 2 synapses, or only one upstream CR neuron were included if their bilateral counterparts (as defined by neuron morphology) were already part of the first selected set. The targets of the 1 st layer in the network were

likewise reconstructed, but only those belonging to the motoneuron class (i.e. cells innervating muscles or ciliary band cells) were included in the circuit. All neurons in the circuit were manually reviewed and weakly connected cells were double checked for accuracy of the synapse annotation. Eight fragments with three or more synapses downstream of the CR neurons but without a cell body were not included in the final circuit. A pair of glial cells, four sensory cells and two interneurons without a bilateral pair were also not included (*Figure 6—source data 1*). The additional penetrating ciliated sensory cells shown in *Figure 6—figure supplement 1* were reconstructed and reviewed only from sensory ending to cell body.

## Phylogenetic reconstruction

In addition to PKD1-1 and PKD2-1, a number of homologs to the PKD1 and TRPP families were found in the *Platynereis* transcriptome. A phylogenetic analysis was carried out in order to resolve their relationships. The amino acid sequences of the three human genes in the TRPP family (TRPP2, TRPP3 and TRPP5) and the five homologs of the PKD1 family (PKD1L1-3, PKDREJ and PC1) were used as queries in a BLAST search for homologs against the NCBI nr database. *Platynereis* PKD1 and PKD2 were also used as queries to find additional sequences. Care was taken to collect sequences from animals across the animal phylogeny. Additional sequences were obtained from the COMPAGEN (*Hemmrich and Bosch, 2008*), and PlanMine (*Brandl et al., 2016*) databases. Full-length sequences were aligned using Clustal Omega (*Sievers et al., 2011*). For the joint PKD1-PKD2 tree, the alignment was cropped to span only the six transmembrane (TM) domains homologous to the TRP-channel homology region common to both families. For the separate PKD2 and PKD1 family phylogenies any clearly alignable region was included. Only full sequences less than 90% identical were used for the alignment. GBlocks alignments (*Talavera and Castresana, 2007*) were used for the phylogenetic reconstruction. Maximum likelihood trees were recovered with PhyML (RRID:SCR_014629; *Guindon et al., 2010*) using the SMS model selection tool (*Lefort et al., 2017*) and aLRT statistics (*Anisimova and Gascuel, 2006*). Sequences were renamed with a custom script (*Bezares-Calderón, 2018*; copy archived at https://github.com/elifesciences-publications/Bezares_et_al_2018).

The GenBank accession numbers of cloned *Platynereis* PKD gene fragments are: PKD2-1, MH035677; PKD2-2, MH035678; PKD1-1, MH035679; LOV-1, MH035680. Full-length sequences from the transcriptome assembly, as well as sequence alignments and the resulting phylogenetic trees are in *Figure 3—figure supplement 1—source data 1*.

## SEM of wild type and mutant nectochaete larvae

Mutant and age-matched wild type larvae were collected by phototaxis and simultaneously processed for SEM analysis. In brief, larvae were fixed in 3% Glutaraldehyde/PBS for 3 days ($PKD2-1^{mut/mut}$ and its wild type control) or for 1 month ($PKD1-1^{i1/i1}$ and its wild type control) at 4°C. Fixative was then washed out overnight in PBS, and samples were postfixed with 1% $OsO_4$ for 2 hr on ice. Samples were gradually dehydrated with EtOH and critical point dried following standard protocols. Samples were sputter coated with an 8 nm layer of platinum (Bal-Tec MED010) and imaged with a Hitachi S-800 field emission scanning electron microscope at an accelerating voltage of 15KV. The genotype of fixed larvae was inferred from their sibling larvae.

## Predation experiments

Adult stages of male and female *C. typicus* were picked individually from culture bottles and transferred to a beaker with fNSW ca. 2 hr prior to the experiments. The experiments were carried out at 18°C in a Ø7 cm beaker completely wrapped with tin foil (i.e. dark conditions) and filled with 200 ml fNSW. 25 phototactic larvae of each genotype (25 age-matched wild type and 25 $PKD2-1^{mut/mut}$) were transferred to the experimental container and then 5 or 10 copepods were added. Experiments were run for 12 hr or 24 hr without shaking (*Almeda et al., 2017*). At the end of the incubation period, live copepods were counted, and every surviving larva was collected for genotyping. We used 12 mutant batches in 42 experiments (duplicates or triplicates with larvae from the same batch were simultaneously run for a given incubation time). In parallel to each experiment, a negative control with 25 larvae of each genotype but without copepods was run under the same environmental settings to assess mortality not related to predation. In the negative controls, 86% or more larvae

survived, while the maximum survival rate with copepods was 70% (Figure 3—source data 4). Predation rates (*I*) for each genotype were calculated according to the following formula

$$I = \frac{C_i - C_f}{NxT}$$

taken from (*Almeda et al., 2017*), where $C_i$ and $C_f$ are initial and final prey concentrations (prey/L), respectively, *N* is number of alive predators at the end of experiment and *T* is incubation time in days. The null hypothesis of equal predation rates was tested against the alternative of a higher predation rate of mutant larvae with a non-parametric one-sided exact Wilcoxon-Pratt signed rank test.

All videos were analysed manually or with custom-written macros in FiJi (RRID:SCR_002285; *Schindelin et al., 2012*), and data plots were generated in R (RRID:SCR_001905). All panels were assembled into figures using Adobe Illustrator CS6 and CC 2018 (Adobe Systems, Inc.).

## Acknowledgments

We thank Ada Kozlowska for help with the initial genotyping of PKD1-1 mutants, Detlev Arendt for providing access to the *Platynereis* genome database, Nadine Randel for fixing samples for SEM, Dorothee Koch and Sinja Mattes for worm culture maintenance, Rocío Rodriguez for copepod culture maintenance and shipments and Elizabeth Williams for comments on the manuscript. The research leading to these results received funding from the European Research Council under the European Union's Seventh Framework Programme (FP7/2007-2013)/European Research Council Grant Agreement 260821. LABC was supported by a grant from the Deutsche Forschungsgemeinschaft (JE 777/3–1). RA was supported by a Marie Curie Intra-European fellowship (6240979) and by the Centre for Ocean Life, a VKR Center of Excellence funded by the VKR Foundation.

## Additional information

### Funding

| Funder | Grant reference number | Author |
| --- | --- | --- |
| Deutsche Forschungsgemeinschaft | JE 777/3-1 | Luis A Bezares-Calderón |
| European Commission | 260821 | Réza Shahidi<br>Gáspár Jékely |
| European Commission | 6240979 | Rodrigo Almeda |
| Max-Planck-Gesellschaft | Open-access funding | Jürgen Berger<br>Sanja Jasek<br>Csaba Verasztó<br>Sara Mendes<br>Martin Gühmann<br>Gáspár Jékely |

The funders had no role in study design, data collection and interpretation, or the decision to submit the work for publication.

### Author contributions

Luis A Bezares-Calderón, Conceptualization, Resources, Data curation, Software, Formal analysis, Validation, Investigation, Visualization, Methodology, Writing—original draft, Writing—review and editing; Jürgen Berger, Data curation, Visualization, Methodology; Sanja Jasek, Data curation, Formal analysis, Maintenance of catmaid server; Csaba Verasztó, Resources, Data curation, Formal analysis, Connectome tracing; Sara Mendes, Data curation, Visualization, Transgenesis; Martin Gühmann, Resources, Data curation, Formal analysis; Rodrigo Almeda, Resources, Data curation, Formal analysis, Centropages typicus culture; Réza Shahidi, Resources, Data curation, Formal analysis, Visualization; Gáspár Jékely, Conceptualization, Data curation, Formal analysis, Supervision, Funding acquisition, Investigation, Methodology, Writing—original draft, Project administration

## Author ORCIDs

Luis A Bezares-Calderón (ID) http://orcid.org/0000-0001-6678-6876
Csaba Verasztó (ID) http://orcid.org/0000-0001-6295-7148
Martin Gühmann (ID) http://orcid.org/0000-0002-4330-0754
Gáspár Jékely (ID) http://orcid.org/0000-0001-8496-9836

## Decision letter and Author response

Decision letter https://doi.org/10.7554/eLife.36262.055
Author response https://doi.org/10.7554/eLife.36262.056

# Additional files

## Supplementary files

• Supplementary file 1. Summary of cells that form part of the CR wiring diagram, proposed function and the evidence supporting it.
DOI: https://doi.org/10.7554/eLife.36262.043

• Transparent reporting form
DOI: https://doi.org/10.7554/eLife.36262.044

## Data availability

Sequencing data have been deposited in Genbank under accession codes MH035677, MH035678, MH035679, MH035680. The neuron reconstructions are available from NeuroMorpho (https://doi.org/10.13021/degz-cz50). Code is available at https://github.com/JekelyLab/Bezares_et_al_2018 (copy archived at https://github.com/elifesciences-publications/Bezares_et_al_2018).

The following datasets were generated:

| Author(s) | Year | Dataset title | Dataset URL | Database and Identifier |
|---|---|---|---|---|
| Luis A Bezares-Calderón, Sanja Jasek, Csaba Verasztó, Sara Mendes, Réza Shahidi, Martin Gühmann, Gáspár Jékely | 2018 | 3D reconstructions of neurons in the *Platynereis* startle circuit | https://doi.org/10.13021/degz-cz50 | NeuroMorpho, 10.13021/degz-cz50 |
| Bezares-Calderón LA, Jékely G | 2018 | Sequencing data | https://www.ncbi.nlm.nih.gov/nuccore/MH035677 | Genbank, MH035677 |
| Bezares-Calderón LA, Jékely G | 2018 | Sequencing data | https://www.ncbi.nlm.nih.gov/nuccore/MH035678 | Genbank, MH035678 |
| Bezares-Calderón LA, Jékely G | 2018 | Sequencing data | https://www.ncbi.nlm.nih.gov/nuccore/MH035679 | Genbank, MH035679 |
| Bezares-Calderón LA, Jékely G | 2018 | Sequencing data | https://www.ncbi.nlm.nih.gov/nuccore/MH035680 | Genbank, MH035680 |

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
