## [Decision Letter]

Thank you for submitting your article "Neural circuitry of a polycystin-mediated hydrodynamic startle response for predator avoidance" for consideration by *eLife*. Your article has been reviewed by Eve Marder as the Senior Editor, a Reviewing Editor, and three reviewers. The following individual involved in review of your submission has agreed to reveal his identity: Ronald L Calabrese (Reviewer #1).

The reviewers have discussed the reviews with one another and the Reviewing Editor has drafted this decision to help you prepare a revised submission.

Summary:

In this manuscript, the authors present compelling data that they have mapped at the connectomic level the complete neuronal circuit for an ecologically relevant startle response in the planktonic larvae of *Platynereis dumerilii*. The experiments involve not only EM reconstruction of the connectome of the circuit, but characterization of the stimulus-response pattern for the startle, morphological and physiological identification of the relevant receptor neurons, genetic demonstration of the necessity of polycystins PKD1-1 and PKD2-1 in the receptor neurons for their function and for the behavioral response, and behavioral/genetic analysis of the relevance of the startle response for avoiding predators. A logical scheme of how the circuit might work is presented. This type of complete description of a behaviorally relevant neuronal network is rare and this example for a startle network is particularly elegant. The paper should arouse general interest in the neurobiological community.

Essential revisions:

1) A significant lapse is in the Discussion section. The authors would do well to comment on Marty Chalfie's classic studies of the escape response circuits of *C. elegans* that stretches from sensory neurons to interneurons to motor neurons and muscle cells. Much is known about the worm startle response pathways from input to output. It is arguably the best studied example in the field but is not mentioned by the authors.

2) Because beyond the receptor neurons the circuit elucidated depends on EM reconstruction, the circuit mechanisms involved in startle are limited to plausible models. Many experiments come to mind that could strengthen the circuit analysis, and if any of these have been done or could be accomplished quickly then they should be included although they are not required.

– Perform the predation assay using *PKD-1* mutants; given PKD-2 has broader neuronal expression, such an experiment may provide evidence to more specifically associate the susceptibility of animals to predation to the absence of CR-dependent startle response (as opposed to functions of other PKD-2-expressing ciliated neurons). This experiment seems like it could indeed be accomplished quickly.

– Perform calcium imaging assay in the *PKD-1/2* mutants, to demonstrate that sensory-evoked CR neural activity depends upon these candidate receptors. The feasibility of this experiment may depend on the ability to put the transgenic calcium reporters into the mutant background.

– Perform genetic rescue of the behavioral mutants: the authors use transheteroallelic combinations, which should alleviate most concerns about phenotypes being due to off-target mutations. However, it would be reassuring to show the startle behavior could be restored by selective expression of PKD in the CR neurons, if such promoter constructs are available.

– Systematically ablate, inactivate, or record downstream interneurons in the network to determine the effect on behavior.

3) No primary EM data is shown. At least one figure or figure supplement showing what the EM level data looks like would be helpful.

---

## [Author Response]

Essential revisions:1) A significant lapse is in the Discussion section. The authors would do well to comment on Marty Chalfie's classic studies of the escape response circuits of C. elegans that stretches from sensory neurons to interneurons to motor neurons and muscle cells. Much is known about the worm startle response pathways from input to output. It is arguably the best studied example in the field but is not mentioned by the authors.

We have rewritten both the Introduction and Discussion section and now refer to the work on the touch withdrawal response in *C. elegans*. We expanded the discussion to compare the *Platynereis* circuit to escape circuits in the nematode and other organisms where startle behaviors have been studied.

2) Because beyond the receptor neurons the circuit elucidated depends on EM reconstruction, the circuit mechanisms involved in startle are limited to plausible models. Many experiments come to mind that could strengthen the circuit analysis, and if any of these have been done or could be accomplished quickly then they should be included although they are not required.– Perform the predation assay using PKD-1 mutants; given PKD-2 has broader neuronal expression, such an experiment may provide evidence to more specifically associate the susceptibility of animals to predation to the absence of CR-dependent startle response (as opposed to functions of other PKD-2-expressing ciliated neurons). This experiment seems like it could indeed be accomplished quickly.– Perform calcium imaging assay in the PKD-1/2 mutants, to demonstrate that sensory-evoked CR neural activity depends upon these candidate receptors. The feasibility of this experiment may depend on the ability to put the transgenic calcium reporters into the mutant background.– Perform genetic rescue of the behavioral mutants: the authors use transheteroallelic combinations, which should alleviate most concerns about phenotypes being due to off-target mutations. However, it would be reassuring to show the startle behavior could be restored by selective expression of PKD in the CR neurons, if such promoter constructs are available.– Systematically ablate, inactivate, or record downstream interneurons in the network to determine the effect on behavior.

We acknowledge the relevance of the experiments proposed, however, we have not been able to carry out these for the present paper due to practical reasons. Our research group recently moved country and we will need time to expand the fragile *PKD* mutant cultures. We did not want to delay the publication of our finding. In the Discussion section, we mention some of these experiments.

Regarding the mutant stocks, we not only tested transheteroallelic combinations, but also did two rounds of outcrossing to reduce the chance of interference from possible off-target mutations.

3) No primary EM data is shown. At least one figure or figure supplement showing what the EM level data looks like would be helpful.

We included a new figure (Figure 6—figure supplement 3) to show examples of the primary EM data, especially focusing on regions where we detected synapses between cells in the circuit.